

# Tidal variability in the Hong Kong region

Adam T. Devlin
*Institute of Space and Earth Information Science, The Chinese University of Hong Kong, Shatin,*
*Hong Kong SAR, China*
Jiayi Pan[*]
*Institute of Space and Earth Information Science, The Chinese University of Hong Kong, Shatin,*
*Hong Kong SAR, China*
*College of Marine Science, Nanjing University of Information Science and Technology, Nanjing,*
*Jiangsu, China*
*Shenzhen Research Institute, The Chinese University of Hong Kong, Shenzhen, Guangdong, China*
Hui Lin
*Institute of Space and Earth Information Science, The Chinese University of Hong Kong, Shatin,*
*Hong Kong SAR, China*
[*] - Corresponding author



**Abstract**

Mean sea-level (MSL) is rising worldwide, and correlated changes in ocean tides are also occurring; their combination may influence future total sea-levels (TSL), possibly increasing coastal inundation and nuisance flooding events in sensitive regions. Analyses of a set of tide gauges in Hong Kong and in the South China Sea (SCS) reveal complex tidal behavior. Most prominent in the results are strong correlations of MSL variability to tidal variability which may further increase local flood levels under future MSL rise. We also highlight inter-tidal correlations of diurnal ($D_1$) tides to semidiurnal ($D_2$) tides, positively reinforced through the northern SCS, and the correlations of overtide (OT) fluctuations to $D_1$ and $D_2$, negatively reinforced (i.e., anti-correlated) across the same region, thought to be related to the baroclinic energetics in the Luzon Strait and the Taiwan Strait. The baroclinic signals may be enhanced at the northern shelf of the SCS and can generate PSI interactions that may amplify minor tides such as $M_3$. Additionally, there are anomalous tidal events observed in some enclosed harbor regions of Hong Kong, corresponding to times of rapidly changing MSL as well as rapid coastal development projects. Results support the hypothesis that the observed variability is due to multiple spatial processes, best described as an amplification of the local (Hong Kong) tidal response to the prevailing regional (SCS) tidal patterns, enhanced by local harbor changes. A close analysis of the full-spectrum tidal response suggests that a change in the resonant and frictional response may have occurred.



## *1. Introduction*

Ocean tides have long been considered to be a stationary process, as they are driven by the gravitational forcing of the Sun and Moon whose motions are complex but highly predictable (Cartwright and Tayler, 1971). Yet, long-term changes in the tides have been observed recently on regional (Ray, 2006; Jay et al., 2009; Zaron and Jay, 2014; Rasheed and Chua, 2014; Feng et al., 2015; Ross et al., 2017) and worldwide spatial scales (Woodworth, 2010; Müller, et al. 2011; Haigh et al., 2014; Mawdsley et al., 2015), concurrent with long-term global mean sea level (MSL) rise (Church and White, 2006; 2011). Since gravitational changes are not the reason, the tidal changes are likely related to terrestrial factors such as: changes in water depth which can alter friction (Arbic et al, 2009), coastal morphology and resonance changes of harbor regions (Cartwright, 1972; Bowen and Gray, 1972; Amin, 1983; Vellinga et al., 2014; Jay et al., 2011; Chernetsky et al., 2010, Familkhalili & Talke, 2016), or stratification changes induced by increased upper-ocean warming (Domingues et al., 2008; Colosi and Munk, 2006; Müller, 2012; Müller et al., 2012), all of which are also related to sea-level rise. Tides can also exhibit short-term variability correlated to short-term fluctuations in MSL. These variabilities may influence extreme water level events, such as storm surge or nuisance flooding (Sweet and Park, 2014; Cherqui et al., 2015; Moftakhari et al., 2015; 2017; Ray and Foster, 2016). Such short-term extreme events are obscured when only considering long-term linear trends. Any significant additional positive correlation between tides and sea-level fluctuations may amplify this effect and implies that flood risk based only on the superposition of present day tides and surge onto a higher baseline sea-level will be inaccurate in many situations. The accurate determination of the nature and impact of sea-level rise and associated tidal change necessitates a regionally- and locally-focused strategy, therefore, analysis of the correlations between tides and sea level can indicate locations where tidal evolution should be considered a substantial modification to sea-level rise. Moreover, since storm surge is a long wave, factors affecting tides can also alter storm surge (Familkhalil and Talke, 2016; Arns et al., 2017), so an improved knowledge of tides can also improve storm response planning and may be instructive in guiding future coastal development.

Recent works have surveyed tidal anomaly correlations (TACs) at multiple locations in the Pacific, examining the sensitivity of tides to sea-level fluctuations (Devlin et al., 2014; Devlin, 2016; Devlin et al., 2017a), finding that over 90% of tide gauges analyzed exhibited some measure of correlation in at least one tidal component. In a related work (Devlin et al.,





2017b), the combined TACs of the four largest tidal components was calculated as a proxy
for the changes in the highest astronomical tide ($\delta$-HAT), with 35% of gauges surveyed
exhibiting a sensitivity of $\delta$-HATs to sea-level fluctuations of at least ±5% in addition to sea-
level change. The greatest $\delta$-HAT response was seen in Hong Kong (+65%), and additional
analyses revealed that TSL exceedance levels have nearly doubled (+150 mm) that of MSL
exceedance alone (+78 mm) over the past 50 years, demonstrating that the non-stationarity of
tides can be a significant contributor to total water levels in this region, and this behavior
warrants closer examination.
*1.1 Sea-level and tides in Hong Kong and the South China Sea*
Hong Kong and the Pearl River Delta (PRD) region contains many densely-populated
urban metropolises with extensive coastal infrastructure, and substantial recent land
reclamation projects. These coastal morphology changes along with sea-level rise may
change the local resonant and frictional response of the local tides to the regional tidal
variability and may contribute to TSL changes and nuisance flooding. Sea-level rise in the
region has exhibited a variable rate in the region over the past 50 years (Li and Mok, 2012),
but a common feature of all sea level records in the SCS is a steep increase in the late 1990s
with a subsequent decrease in the early 2000s, then followed by a sustained increase to the
present day. In addition to the variable MSL behavior, there are also anomalous tidal events
observed at gauges in semi-enclosed harbor regions during the late 1990s and early 2000s
(shown and discussed below), corresponding to times of both rapidly changing sea level and
aggressive land reclamation.
Understanding the tidal behavior in Hong Kong requires a thorough examination of
the tidal dynamics in the South China Sea. Both diurnal ($D_1$) and semidiurnal ($D_2$) tides enter
the SCS from the Pacific through the Luzon Strait. The $D_2$ constituents are damped by a
factor of two as they enter the SCS, and the $D_1$ constituents are amplified by a similar factor
(Zu et al., 2008; Fang et al., 1999; Jan et al., 2007). The semidiurnal wave bifurcates,
partially travelling northwest towards the Taiwan Strait, and partially travelling southwest
towards the Sunda Shelf, though the diurnal wave only propagates southwest. The part of the
semidiurnal wave that travels towards the Taiwan Strait meets the large incoming semidiurnal
energy from the East China Sea (ECS). The semidiurnal tides have very large amplitudes (~
2m) here and exhibit a $D_2$ resonance on the western side of the Strait via a partial quarter-



wave resonance (Jan et al., 2004).  In addition, a large amount of $D_2$ internal energy is
generated, though little to no $D_1$ baroclinic energy is observed.

The Luzon region is one of the most active regions of baroclinic generation in the

world ocean (Wang, 2012). Approximately one third of the $K_1$ surface tide energy (~ 12 GW)
is converted to baroclinic energy (Jan et al. 2007), and about one quarter of the $M_2$ surface
tide is converted to the baroclinic tide (Niwa and Hibiya, 2004). The surface tide expression
in the SCS is dependent on the baroclinic conversion, which is in turn highly sensitive to the
geometric and environmental properties of the Luzon Strait (Jan et al., 2008; Wang, 2012).
Internal tides yield a high-mode vertical velocity structure that tends to dissipate tidal energy
close to the generation site as well as a low-mode energy that can travel for thousands of
kilometers (Liu et al., 2015). Therefore, even at a great distance from the generation site,
much of the baroclinic energy may remain coherent. Internal tides can propagate as narrow
beams, which may be enhanced upon arrival at the shelf (Lien et al., 2005), and nonlinear
interactions are enhanced within the tidal beams, in areas where internal tide beams are
reflected (Mercier et al., 2012), or in regions where the tidal beams intersect (Teoh et al.,
1997; Korobov and Lamb, 2008).

The $D_1$ and $D_2$ internal tides may interact with each other as well as with other

frequencies, such as the local inertial frequency, $f$, via parametric subharmonic instability
(PSI) interactions (McComas and Bretherton, 1977; MacKinnon and Winters, 2005), a form
of resonant triad interactions (Craik, 1985).  Previously, such interactions were only thought
to occur near the critical latitude (~29° for $M_2$) where $f$ is equal to half the $M_2$ frequency (see
e.g., Alford, 2008).  However, for the case where a PSI interaction turns from weakly
nonlinear to strongly nonlinear, it can enhance generation at subharmonics different from
exactly half the frequency (Korobov and Lamb, 2008).  For example, the presence of a
resonant triad interaction between $M_2$ and $K_1*O_1$ was observed in the Solomon Sea (Devlin et
al., 2014). Many PSI-type tidal interactions have been observed in the SCS. Kinetic energy
spectra from a current profiler on the northern continental slope near Dongsha Island (~20°
N), halfway between the Luzon Strait and Hong Kong) revealed strong peaks at the nonlinear
interaction frequencies of $fM_1$ ($f + M_1$) and $M_3$ ($M_1 + M_2$), (Xie et al., 2008) as well as other
components in the $D_3$ band (e.g., $MO_3$).  The presence of the PSI interactions was confirmed
by bicoherence estimates (Carter and Gregg, 2006), and validates previous suggestions that
PSI interactions can occur equatorward of the critical latitudes depending on stratification and



circulation conditions (Xie et al, 2011).  Other PSI interactions were observed in the southern
regions of the SCS (Chinn, 2012; Liu, 2015).

*1.2 Outline of this study*

It is hypothesized that the observed tidal variability in Hong Kong is due to either: 1)

regional changes in the dynamics of the SCS such as MSL rise, circulation patterns, or upper-
ocean warming and stratification, 2) local changes in friction and/or resonance related to land
reclamation projects, or 3) a combination of coupled mechanisms at multiple spatial and
temporal scales.  To determine the relevant scales of variability, we perform a spatial and
temporal analysis of tidal sensitivity to MSL variations in Hong Kong and the SCS.  This
manuscript is structured as follows.  After the introduction, the data inventory will be
described, and a description of the TAC and δ-HAT methods will be given.  Following this
will be the results, detailing the spatial and temporal patterns of the TAC and δ-HAT
determinations. We will then closely examine extreme tidal anomalies in Hong Kong by
analyzing the full tidal response, including minor tidal components, and will compare
regional correlations of tidal properties in the historical and modern eras.   Following the
results, a discussion of relevant spatial scales and mechanisms is presented, as well as future
proposed works.
**2. Methods**

*2.1 Data sources*

A set of 13 tide gauges in the Hong Kong region were provided by the Hong Kong

Observatory (HKO) and the Hong Kong Marine Department (HKMD).  The longest record is
the North Point/Quarry Bay (QB) tide gauge, located in Victoria Harbor.  The gauge was
established in 1954 and was relocated from North Point to Quarry Bay in 1986, and the
datums were adjusted and quality controlled by HKO to provide a continuous record (Ip and
Wai, 1990).  Five more gauges are provided by HKO: Tsim Bei Tsui (TBT; 1974-present),
Tai Po Kau (TPK; 1963-present), Shek Pik (SP; 1999-presnt), Tai Miu Wan (TMW; 1996-
present), and Waglan Island (WAG; 1995-present).  In addition, four locations are operated
by the HKMD (Cheung Chau; CHC, Kwai Chung; KC, Ma Wan; MW, and Ko Lau Wan;
KLW); all have been recording from ~2004 to the present day.  Next, there are some
additional data records originally operated by HKO that are no longer active (Chi Ma Wan
(CMW; 1963-1997) and Lok On Pai (LOP; 1981-1999)).  Additionally, historical data from
China in the Beibu Gulf and the Taiwan Strait are downloaded from the University of Hawaii




Sea Level Center (UHSLC; website): Shanwei (SW), Zhapo (ZP), Beihei (BH), Haikou
(HK), Dongfang (DF), and Xiamen (XM).  These data records are all continuous from 1976-
1997, except for Xiamen which runs from 1954-1997.  Rounding out this inventory are six
other locations in the SCS acquired from UHSLC; Manila (MN) in the Philippines (1984-
2016), Kaohsiung (KS) and Keelung (KL) in Taiwan (1980-2014), Vung Tau (VT), Vietnam
(1986-2002; 2007-2014); Sedili (SD), Malaysia (1986-2016), Bintulu (BT), Malaysia (1992-
2016), and one location on the outside of the SCS closest to the Luzon Strait (Ishigaki Island;
IG, 1968-2013) to provide a comparison to the tides within the SCS.  The TACs and δ-HATs
at these last seven locations were already reported on in Devlin et al. (2017a; 2017b), here,
they are recalculated with updated data to compare the spatial coherence of tidal dynamics in
the SCS to Hong Kong. Gauge locations in Hong Kong are shown in Figure 1, with the
gauges from HKO indicated by green markers, gauges from HKMD by light blue, and
historical (non-operational) gauges by red. SCS gauges are shown in Figure 2; green indicates
gauges that are actively updated, red indicates gauges that have not been updated since 1997.
Table 1 lists the metadata for all locations, including latitude, longitude, and record length.
*2.2 Tidal admittance calculations*

Investigations of tidal behavior rely on a tidal admittance method. An admittance is

the unitless ratio of an observed tidal constituent to the corresponding tidal constituent in the
astronomical tide generating force expressed as a potential, *V*, divided by the acceleration due
to gravity, *g*, to yield $Z_{pot}(t) = V/g$, with units of length that can be compared to tidal
elevations, $Z_{obs}(t)$, via harmonic analysis.  Yearly harmonic analyses are performed on both
$Z_{obs}(t)$ and $Z_{pot}(t)$ at each location, using the R_T_TIDE package for MATLAB (Leffler and
Jay, 2009), a robust analysis suite based on T_TIDE (Pawlowicz, 2002). Because nodal and
other low-frequency astronomical variabilities are present with similar strengths in both the
observed tidal record and in $Z_{pot}(t)$, their effects are eliminated in yearly analyzed admittance
time series. The tidal potential is determined based on the methods of Cartwright and Tayler
(1971). The result from a single harmonic analysis of $Z_{obs}(t)$ or $Z_{pot}(t)$ determines an
amplitude, *A*, and phase, *θ*, at the central time of the analysis window for each tidal
constituent, with error estimates.  A moving analysis window produces time-series of
amplitude, *A(t)*, and phase, *θ(t)*, with the complex amplitude, **Z**(t), given by:

$$\mathbf{Z}(t) = A(t)e^{i\theta(t)}.$$

(1)





The tidal admittance (**A**) and phase lag (**P**) are formed using Eqs. (2) and (3)
$$\mathbf{A}(t) = abs\,|\frac{\mathbf{Z}_{obs}(t)}{\mathbf{Z}_{pot}(t)}|\ , \tag{2}$$

$$\mathbf{P}(t) = \theta_{obs}(t) - \theta_{pot}(t)\ . \tag{3}$$

The harmonic analysis procedure also provides an MSL time-series.  For each resultant
dataset (MSL, **A** and **P**), the mean and trend are removed from the time series to allow direct
comparison of their co-variability.  The magnitude of the long-term trends is typically much
less than the magnitude of the short-term variability, which is now more apparent in the data
(Devlin et al., 2017a).

Tidal sensitivity to sea-level fluctuations is quantified using tidal anomaly correlations

(TACs), the relationships of detrended tidal variability to detrended MSL variability. We
determine the sensitivity of the amplitude and phase of individual constituents ($M_2$, $S_2$, $K_1$,
$O_1$, $N_2$, $K_2$, $P_1$, and $Q_1$) to sea-level perturbations at the yearly-analyzed scale. We also
consider the change in the highest astronomical tide (δ-HAT), estimated in two ways. First,
by combining the yearly analyzed time series of the four largest tidal amplitudes ($M_2$, $S_2$, $K_1$,
and $O_1$), approximately 75% of the full tidal height (δ-$HAT_4$), and secondly by considering
the combination of all eight constituents, approximately 95% of the full tidal height (δ-
$HAT_8$). The latter determination provides a better approximation to the full tidal range,
though the former provides a more statistically stable value, as the four minor constituents are
more prone to noise and spurious fluctuations. The detrended time series of the δ-HATs are
compared to detrended MSL variability in an identical manner as the TACs, and both are
expressed in units of millimeter change in tidal amplitude per 1-meter fluctuation in sea-level.
These units are adopted for convenience, though in practice, the observed fluctuations in
MSL are on the order of ~ 0.25 m. The phase TACs are reported in units of degree change
per 1-meter fluctuation in sea-level.

The TAC methodology can also be used to examine correlations between different

parts of the tidal spectrum. We also consider the sensitivity of combined diurnal ($D_1$; $K_1 + O_1$
$+ P_1 + Q_1$) tidal perturbations to semidiurnal ($D_2$; $M_2 + S_2 + N_2 + K_2$) tidal perturbations
($D_1/D_2$ TACs).  Additionally, we calculate the sensitivity of tidal range to frictional changes,
by considering the combined variations of the seven largest overtides (OT; $M_4$, $M_6$, $S_4$, $MK_3$,
$MO_3$, $SN_4$, and $MN_4$), to fluctuations in the combined $D_1$ and $D_2$ amplitude (OT TACs).  The
units of the $D_1/D_2$ and OT TACs are dimensionless (i.e., mm/mm), and statistics are



calculated as above. We assume that the interannual variability captured by all TACs and δ-
HATs can be extrapolated to longer time scales, subject to the qualification that the changes
remain "small-amplitude", i.e., a 0.5 to 1m change in MSL and a change in tidal amplitude of
a few 10s of cm.
The definition of the year window used for harmonic analysis may have an influence
on the value of the TAC or δ-HAT, e.g. calendar year (Jan-Dec) vs. water year (Oct-Sep). To
provide a better estimate of the overall correlations for all data we take a set of
determinations of the correlations using twelve distinct year definitions (i.e., one-year
windows running from Jan-Dec, Feb-Jan, …, Dec-Jan.). We take the average of the set of
significant determinations (i.e., $p$-values of $< 0.05$) as the magnitude of the TAC or δ-HAT.
For an estimate of the confidence interval of the TAC or δ-HAT, the interquartile range
(middle 50% of the set) is used. A step-by-step description of the TAC and δ-HAT methods,
including the details of the calculations of the regressions and statistics can be found in the
supplementary materials of Devlin et al. (2017b). For the very long record stations (e.g., QB
and TPK), we only consider the past 30 years for TAC and δ-HAT determinations, and for
other stations, we use the full record, though some locations are less than 30 years, and some
are historical.
We also highlight some anomalous tidal events observed around the turn of the
century at certain Hong Kong gauges, and we compare and discuss the coherence and
evolution of the tidal behavior in Hong Kong and the SCS via correlation analysis. We
consider the eight tides in the $D_1$ and $D_2$ band, as well as the $2N_2$, $M_3$ and $MO_3$ tides (for
reasons that will be made clear later), and MSL. All gauges are compared to the Quarry Bay
gauge as the "standard", and we consider a demarcation time between "historical" and
"modern" as 1997. For the early record, we use the Hong Kong data at CMW, TPK, LOP,
and TBT, the historical data from the mainland of China (to represent the historical SCS).
For the modern era, we consider all operational data in Hong Kong, as well as Manila, Vung
Tau, Sedili, Ishigaki, and the two Taiwan gauges. We use Ishigaki to represent the Pacific in
both time periods. For all comparisons, we only use the data that overlaps the QB record.
Due to the nature of the time coverage at our set of gauges, only two gauges will allow a
direct comparison in both time periods in Hong Kong (TPK and TBT). However, a few other
locations in the historical and modern sub-sets are located close enough to each other to allow
a near-direct pairing; Lok On Pai/Ma Wan, and Chi Ma Wan/Cheung Chau.





### 3. Results

The individual TACs for amplitude and phase in Hong Kong and the SCS are
discussed first, followed by the δ-HATs, the $D_1/D_2$ TACs, and the OT TACs.  In all figures,
significant positive results will be reported by red markers, significant negative results by
blue markers, and insignificant values are shown as black markers. The relative size of the
markers will indicate the relative magnitude of the TAC or δ-HAT according the legend scale
on each plot.  All numerical results for the major amplitude TACs ($M_2$, $S_2$, $K_1$, and $O_1$) are
listed in Table 2, and the δ-HATs, $D_1/D_2$ TACs, and the OT/ ($D_1 + D_2$) TACs are listed in
Table 3. Phase TACs of the major constituents, minor constituent ($N_2$, $K_2$, $P_1$, $Q_1$) amplitude
TACs, and the other OT TACs (i.e., OT/$D_1$ and OT/$D_2$) are reported in Table S1, S2 and S3
of the supplementary material.  Phase TACs for the minor constituents are insignificant at all
locations and are not reported or plotted.  Next, we explore the anomalous tidal events seen at
Hong Kong gauges in recent years by analyzing the behavior of major and minor tidal
components. Finally, we compare correlations between early and later eras to explore the
temporal coherency of tidal behavior.

### 3.1 Tidal anomaly correlations (TACs)

We first show the semidiurnal TACs in Hong Kong (Figure 3 (a) and (c)) and in the
SCS (Figure 3 (b) and (d)).  In Hong Kong (Fig 3(a)), the strongest positive $M_2$ TACs are
seen at Quarry Bay ($+218 \pm 37$ mm m$^{-1}$), and at Tai Po Kau ($+267 \pm 42$ mm m$^{-1}$), with a
smaller positive TAC seen at Shek Pik.   In the waters west of Victoria Harbor, all gauges
except Kwai Chung exhibit moderate negative TACs. In the SCS (Fig 3(b)), very large and
positive TACs are seen at the three stations in the Beibu Gulf (Dongfang, Beihei, and
Haikou), with values of $+190$, $+460$, and $+379$ mm m$^{-1}$, respectively.  The semidiurnal phase
TACs in Hong Kong (shown in the Supplementary materials, Figure S1(a)) show an earlier
$M_2$ tide under higher MSL at QB and TPK ($-15 \pm 2$ and $-28 \pm 6$ deg m$^{-1}$, respectively), and a
later tide west of Victoria Harbor.  In the SCS (Fig S1(b)), later tides are observed at Manila,
Kaohsiung, and Shanwei, while earlier tides are seen in the Beibu Gulf and at Xiamen.  The
$S_2$ results in Hong Kong (Fig 3(c)) reveal that only QB and TPK have significant amplitude
TAC values (though smaller than $M_2$), and the rest of the SCS has a nearly identical spatial
distribution as $M_2$ (Fig 3(d)).  The $S_2$ phase TACs in Hong Kong (Figure S1(c)) again show
an earlier tide at QB and TPK under higher MSL, and results in the SCS (Figure S1(d)) are
also similar to $M_2$.  The minor semidiurnal amplitude TACs are mainly insignificant in Hong

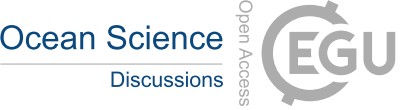



Kong, though $N_2$ has a significant positive TAC at TPK of $+85 \pm 12$ mm m$^{-1}$ (Figure S2(a)),
and $K_2$ has a small significant TAC at both TPK and QB (Figure S2(c)). In the SCS, all TACs
are insignificant or small for $N_2$ (Figure S2(b)), but the $K_2$ response in the Beibu Gulf gauges
is exceptionally large ($+67$ to $+175$ mm m$^{-1}$), notable for such a small-magnitude constituent
(Figure S2(d)).

The diurnal TACs in HK and the SCS generally exhibit a larger-magnitude and more

spatially-coherent response than semidiurnal TACs (Figure 4). Like $M_2$, the strongest $K_1$
values in Hong Kong (Fig 4(a)) are seen at QB ($+220 \pm 15$ mm m$^{-1}$) and TPK ($+190 \pm 68$ mm
m$^{-1}$). In the SCS, the largest magnitude TACs are again found in the Beibu Gulf ($+180$ to
$+578$ mm m$^{-1}$), but unlike $M_2$, all significant TACs are positive in the region (Fig 4(b)), and
there is a significantly large TAC at Bintulu. The $O_1$ results in Hong Kong (Fig 4(c)) and in
the SCS (Fig 4(d)) are like the $M_2$ results, showing positive TACs at QB ($+146 \pm 11$ mm m$^{-1}$)
and TPK ($+100 \pm 25$ mm m$^{-1}$), and strongly negative TACs west of QB. The $O_1$ response in
the SCS is very similar to $K_1$, though a negative response is now seen at Xiamen and
Shanwei, and a small positive response is seen at Keelung. Phase TACs for $K_1$ are mainly
insignificant in Hong Kong (Figure S3(a)), and $O_1$ phase TACs (Figure S3(c)) are only
significant at QB. In the SCS, strong positive phase TACs are seen at Shanwei and
Kaohsiung in both $K_1$ (Figure S3(b)) and $O_1$ (Figure S3(d)), and negative phase TACs for $K_1$
and $O_1$ are seen in the Beibu Gulf. The minor $P_1$ tide has a positive TAC at QB and Ma Wan
($+71 \pm 10$ and $+65 \pm 9$ mm m$^{-1}$; Figure S4(a)), and results are coherent throughout the rest of
the SCS, with positive responses seen in the Beibu Gulf of $+50$ to $+153$ mm m$^{-1}$, and all
other locations having negative responses of $-19$ to $-55$ mm m$^{-}$(Fig S4(b)). The results for $Q_1$
are mixed in Hong Kong (Figure S4(c)), with a positive TAC at QB, a negative TAC at Kwai
Chung and Chi Ma Wan. The $Q_1$ TACs are insignificant at all stations in the SCS (Figure
S4(d)).

*3.2 Change in the highest astronomical tide (δ-HAT)*

The TACs are widely observed in Hong Kong and across the SCS. Conversely, the δ-

HATs (Figure 5) are only of significance at discrete locations. In Hong Kong, five stations
exhibit significant δ-HAT$_4$ values (Fig 5(a)), with QB and TPK having very large positive
magnitudes ($+665 \pm 85$ mm m$^{-1}$ and $+612 \pm 210$ mm m$^{-1}$, respectively), and Shek Pik having
a lesser magnitude of $+138 \pm 47$ mm m$^{-1}$. Conversely, Ma Wan and Chi Ma Wan exhibit
moderate negative δ-HAT$_4$ values, ($\sim -100$ mm m$^{-1}$). The same five gauges are significant for



the δ-HAT$_8$ determinations (Fig 5(c)), though the overall magnitudes are larger (e.g., +834 ±
108 mm m$^{-1}$ at QB and +797 ± 139 mm m$^{-1}$ at TPK).  In the SCS, the δ-HAT$_4$ determinations
are extraordinarily large in the Beibu Gulf, with magnitudes of +813 to +1405 mm m$^{-1}$
(Figure 5(b)), and the δ-HAT$_8$ values are even larger; ~ 20% larger at Haikou and Dongfang,
and at Beihei, nearly 60% larger, showing a positive change in tidal range of > 2 meters for a
1-meter sea-level fluctuation (Figure 5(d)).  Elsewhere in the SCS of note, there are very
large δ-HAT values seen at Bintulu, though this is mostly due to the very large D$_1$ TACs; the
D$_2$ band contributes very little to the change in tidal range here.
*3.3 D$_1$/D$_2$ TACs and OT TACs*
The D$_1$/D$_2$ and OT TACs are important in the northern SCS and are less significant in
the southern reaches. In Hong Kong, all significant D$_1$/D$_2$ TACs results are positive (Figure
6(a)), and at most locations the correspondence is nearly 1-to-1 (e.g., QB; +1.08 ± 0.05, TPK;
+1.01 ± 0.04, TMW; +1.04 ± 0.20), indicating that a change in D$_1$ can yield a nearly-identical
magnitude change in D$_2$, and vice-versa.  Smaller magnitude relations are seen in the western
areas of the domain (e.g., TBT, +0.37 ± 0.02 and LOP; +0.26 ± 0.05). In the SCS (Figure
6(b)), the strongest relationships are in the Beibu Gulf.  At Beihei, the value is nearly 1-to-1
(+1.22 ± 0.03), but at Dongfang, the response is significantly larger than 1 (+2.86 ± 0.19),
and at Haikou, the response is less than 1 (+0.61 ± 0.05).  Elsewhere, small negative relations
are observed near the Taiwan Strait, and large negative relations are seen in the southern
SCS.
The OT TACs at half of gauges in Hong Kong (Fig 6(c)) and nearly every gauge in
the northern SCS (Fig 6(d)) are significant and negatively correlated.  Friction is expected to
be important in coastal or harbor regions, and indeed, the strongest correlations are found in
semi-enclosed or partially protected areas (e.g., QB and Kwai Chung in and near Victoria
Harbor, Tsim Bei Tsui in Shenzhen Bay and TPK in Tolo Harbor).  The largest OT TAC in
Hong Kong is -3.62 ± 0.99 at QB, meaning that for a negative change in the OT component
(which would indicate a reduction of friction) of 1 mm, an increase of 3.62 mm will be seen
in the forcing tides.   In the SCS, the largest (-5.10 ± 0.15) response is seen at Beihei near the
end of the Beibu Gulf. The southern parts of the SCS show no significant relations.  The OT
variability was also compared to the D$_1$ and D$_2$ bands individually, shown in the
supplementary materials (Figure S5), showing that the D$_2$/OT relations are generally more
coherent.





*3.4 Anomalous tidal events in Hong Kong*

We now examine the temporal behavior of the tides in Hong Kong. In Figure 7, the time series of water level spectrum components are shown for QB and TPK, presenting the $D_1$ band (a), the $D_2$ band (b), the OT band (c) and mean sea-level (MSL) (d), given as normalized amplitudes with mean values shown in the legends. Some very notable features of these records are clear. At QB, the early part of the record shows nearly constant tidal amplitudes in $D_1$, while $D_2$ amplitudes show a slight decrease, and MSL exhibits a slight positive trend. In the mid-1980s, however, both $D_1$ and $D_2$ increase drastically until around the year 2003, at which time both tidal bands undergo a rapid decrease of amplitude of ~15%, sustaining this diminished magnitude for about five years before increasing nearly as rapidly. The OT band shows a sustained increase over the historical record, but many of the fluctuations around the trend are anti-correlated to the perturbations in $D_1$ and $D_2$, and during the times of diminished major tides, the OTs increase by about +20%. The MSL record is also highly variable at QB, with a nearly flat trend during the increase in tides seen in the 1980s, followed by a strong increase from ~1993-2000, and then a steep decrease concurrent with the time of diminished tides before increasing again. The gauge at TPK shows a similar tidal behavior, though timings and magnitudes are different here. The increase in $D_1$ and $D_2$ at TPK in the 1980s is much larger and peaks earlier than QB, reaching a maximum around 1996, and then decreasing around 1998, about five years before the drop at QB. Both locations experience an absolute minimum around 2007 in $D_2$, but the $D_1$ minimum at TPK leads the QB minimum by a few years.

We now examine whether these anomalous events are also apparent at other locations in Hong Kong. In Figure 8, the detrended $D_2$ variability of all gauges is presented as normalized amplitudes. The longest record gauges (QB and TPK) displayed in Figure 7 are shown as heavy lines (blue and red, respectively), with the other gauges shown as thinner lines according to the legend. Horizontal lines indicate a change of ± 5% from the mean. At QB and TPK, the variability during the anomaly is 10-15% of the mean, but such a large anomaly is not clearly apparent elsewhere, and most other gauges show a variation of only a few percent. There does appears to be a similar pattern suggested at TBT, with an increase from ~1988 to 1995, a decrease until 2007, and an increase afterwards; however, this gauge has some large data gaps during this time, so a confident determination of the tidal behavior is unlikely without more observations. Very similar results are seen when considering the $D_1$



band, shown in the Supplementary material (Figure S6), as well as for the $M_2$ and $K_1$
amplitudes (Figures S7 and S8).

### *3.5 Minor constituent behavior*

These anomalies in tidal amplitudes are curious by themselves, however, looking at
minor constituents reveals more interesting details. In Figure 9, we present some minor tidal
variability as normalized amplitudes for a selection of representative Hong Kong gauges
(QB, TPK, TBT, CMW, TMW, MW). The $N_2$ amplitudes at all Hong Kong stations exhibit a
long-period harmonic signal, in phase at all locations, corresponding to the lunar eccentricity
cycle of 8.85 years (Fig 9(a)). Typically, this longer-cycle component of the gravitational
potential is suppressed in the admittance analyses, but if there is any terrestrial amplification
of the $N_2$ signal, it may be apparent in the post-admittance analyses. There are regular
maxima starting from the beginning of the record up to ~2002 at which time a minimum of
the cycle is "missed", with the next subsequent minimum being more extreme than all
previous minima. This event corresponds with the major anomaly seen in all constituents at
QB and TPK. More interestingly, the $N_2$ signals at Hong Kong tide gauges are all in phase,
with a near-simultaneous minimum around 2009. The $2N_2$ tide has a similar gravitational
origin as $N_2$ (Fig 9(b)) and exhibits a similar long-period harmonic signal of ~ 4.425 year
(8.85/2 year). Before the anomaly period, the $2N_2$ signal is relatively uncorrelated and noisy,
but after ~2000, the spatial coherence of $2N_2$ increases, while the $N_2$ coherency decreases.
After 2009, the harmonic signal is no longer evident in $N_2$, as there is no clear maximum in
~2013. The $M_3$ tide, usually small (<5 mm) and noisy in the ocean, is significant at all Hong
Kong gauges (~15-25 mm), and also exhibits a ~ 8.85-year signal at all gauges (Fig 9(c)).
There is again a large anomaly present at all gauges after the turn of the 21$^{st}$ century, though
the $M_3$ minimum leads the $N_2$ minimum by a few years due to a phase shift. Another
component of the $D_3$ spectrum, the $MO_3$ tide, also displays a coherent 8.85-year signal (Fig
9(d)). This tidal constituent is typically thought of as a shallow-water overtide but can also
arise via nonlinear interactions between $M_2$ and $O_1$.
The spatial coherence of the minor tides is not as clear in the greater SCS. Figure 10
displays the same constituents at selected gauges in the SCS. We use Quarry Bay again (to
represent Hong Kong), Xiamen (to represent the Taiwan Strait), Dongfang (to represent the
Beibu Gulf), Vung Tau (to represent the central SCS), Sedili (to represent the Gulf of
Thailand) and Ishigaki (to represent the Pacific Ocean). The $N_2$ tide is very strong within the



Taiwan Strait (~350 mm at Xiamen), and of moderate amplitude elsewhere (Fig 10(a)). The
long-period harmonic signal is also present at most gauges with a similar relative variability,
though Dongfang is more variable and noisy, and no other locations shows such a large
relative anomaly as QB circa 2009. The $2N_2$ tide is less coherent regionally than Hong Kong
(Fig 10(b)), though the correlations between Vung Tau and Sedili do appear to be slightly
better after ~2000. At Xiamen, $2N_2$ has the largest observed magnitude (~ 50 mm), and the
~4.425 yr signal is strong, but opposite in phase to QB. For $M_3$, the long-period signal is
generally not observed to be strong in areas of the SCS away from Hong Kong. However,
Xiamen does show a large relatively variable signal, which, like $2N_2$, is opposed in phase to
QB. Finally, the $MO_3$ tide (Fig 10(d)) does not appear to be important away from Hong
Kong; there is a signal suggested at Ishigaki, but the mean value is very small (~ 3 mm), and
this may be attributed to noise.

*3.6 Early correlations vs. modern correlations*

From looking at Figures 7 through 10, it is apparent that there is more variability in

the later years of the record than in the earlier parts of the record. This suggests the
possibility of a recent regime change in the tidal behavior in the Hong Kong and SCS and
warrants a closer examination. We compare the correlations of QB with other gauges in
Hong Kong and the SCS for both the "historical" and "modern" data sets described above to
determine the relevant spatial and temporal scales of tidal variability, including the minor
constituents considered in Figures 9 and 10. Correlation values for $M_2$, $K_1$, $M_3$, $MO_3$, $N_2$, and
$2N_2$ amplitudes are given in Table 4. Table S3 gives the correlations for $S_2$, $O_1$, $K_2$, $P_1$, $Q_1$,
and MSL. Table entries give two entries for longer gauges who cover both time periods (e.g.,
QB, TPK, and IG), as well as a few station pairs that are close enough geographically to
allow a direct comparison (CMW/CHC and LOP/MW), separated by a "/". Gauges that do
not have data during either period will be indicated by a "~". Additionally, the average
correlation at all gauges in HK and the SCS are given for both eras, and the better correlation
between eras will be indicated by bold text.

Results show that the tidal correlations in the region are generally less significant in

the later record than the early record. At Tai Po Kau, all constituents have a strong
correlation in early years (+0.63 to +0.83) but show a lesser correlation in later years (+0.16
to +0.60). At Tsim Bei Tsui, however, the correlation is somewhat better in later years for
semidiurnal constituents. The comparison of Lok On Pai to Ma Wan shows lesser





correlations in later years (+0.06 to +0.76) than in early years (+0.35 to +0.87), and the same
situation is seen when comparing Cheung Chau (+0.02 to +0.61) to Chi Ma Wan (+0.34 to
+0.69). The average correlations of Hong Kong gauges are lower in later years than in early
years; e.g. the $M_2$ average correlation decreases from +0.62 to +0.28, and $K_1$ from +0.54 to
+0.31. In the SCS, historical $M_2$ and $K_1$ average correlations are +0.45 and +0.48, but the
modern correlations are much smaller (both ~ +0.17). The $N_2$ tide is highly correlated to QB
in both time periods at nearly all gauges in HK and the SCS, due to the long-period harmonic
discussed above, but these correlations have decreased from +0.75 in HK and +0.66 in the
SCS to +0.67 and +0.48, respectively. The exception to the pattern of decreasing
correlations is the $2N_2$ tide, whose correlations increase in the modern era (+0.59 to +0.86 in
HK and +0.29 to 0.41 in the SCS). The $M_3$ tide is highly correlated to QB at most HK
gauges (+0.75 to +0.90) which shows similar correlations in both eras; but in the SCS, the $M_3$
correlations are only strong near HK at Zhapo, Shawei, and Xiamen (though at Xiamen, the
tide is anti-correlated to QB). Finally, The $MO_3$ tide is highly correlated at all locations in
HK (+0.78 to +0.92), having increased slightly in the modern era, but in the SCS is only
important very near to HK (Zhapo and Shanwei). These correlation changes confirm what
was suggested by Figure 9 and 10.
***4. Discussion***

*4.1 Spatial scales of tidal variability*

This survey has identified several varieties of tidal variability in Hong Kong and the

SCS that suggest multiple spatial scales of importance. The TAC (Figures 3 and 4) and δ-
HAT (Figure 5) results appear to be more important on a local basis, as the strongest
responses are mainly concentrated at specific locations (e.g., The Beibu Gulf, QB and TPK).
These locations also have significant positive correlations of the four largest tidal amplitudes
to a positive MSL fluctuation, and both locations show a negative response (earlier arriving
tide) of semidiurnal tidal phases. Other locations show a mixed result. The $M_2$ response is
negative at gauges just west of QB (CHC, CMW, MW) and positive at SP, with a similar
pattern seen for the $O_1$ and $Q_1$ amplitude TACs. Conversely, the $K_1$ TAC results are generally
positive. Minor constituent TACs are generally unimportant in Hong Kong, but TPK is more
sensitive to the semidiurnal minor tides, while QB tends to be more sensitive to the diurnal
band. At both QB and TPK, the positive reinforcements of individual tidal fluctuations lead
to very large δ-$HAT_4$ and δ-$HAT_8$ values, though large negative δ-$HAT_4$ and δ-$HAT_8$ values





are seen near to QB at CMW and MW.  The spatial connections in the semi-enclosed center
harbor regions suggest a connected mechanism; this area is where the majority of recent
Hong Kong coastal reclamation projects have occurred, including the construction of a new
island for an airport, shipping channel deepening and other coastal morphology changes.
Such changes in water depth and coastal geometry strongly suggest a relation to frictional or
resonance changes.  The TACs in the Beibu Gulf are strongly positive for most constituents,
and the δ-HATs are even larger than those seen at Hong Kong. Away from Hong Kong and
the northern SCS, TAC and δ-HATs are of less significance.
The $D_1/D_2$ TAC relations (Figures 6 (a) and (b)) are a more regionally-relevant
phenomenon, being significant nearly everywhere in Hong Kong and in the northwest and
north-central SCS, and less significant in the Taiwan Strait and the southern SCS. The
majority of significant $D_1/D_2$ TACs are positive, with most being nearly 1-to-1 (i.e., a ~1-mm
change in $D_1$ will yield a ~1-mm change in $D_2$), confirmed by the close similarity of tidal
behavior of the $D_1$ and $D_2$ tidal bands in Hong Kong (e.g., Figure 7 and Figure 8).  This
aspect of tidal variability in Hong Kong is likely related to the dynamics near the Luzon
Strait, where large amounts of baroclinic conversion in both $D_1$ and $D_2$ tides tend to couple
the variabilities (Jan et al., 2007; 2008; Lien et al., 2015).  The low-mode baroclinic energy
can travel great distances, being enhanced upon arrival at the shelf and leading to the further
generation of energy at non-traditional frequencies such as $f$ and $M_3$ (Xie et al., 2008; 2011;

2013).

There are two sub-regional exceptions to the $D_1/D_2$ correlations.  First, the western
part of Hong Kong the relationships are markedly less than 1 to 1 (~0.33 to ~0.25 at TBT and
LOP, respectively). This may be partially influenced by effects of the Pearl River, which
discharges part of its flow along the Lantau Channel.  The flow of the river is highly seasonal
and ejects a freshwater plume at every ebb tide that varies by prevailing wind conditions and
by the spring-neap cycle (Gu et al., 2012; Pan et al, 2014).  The plumes may affect turbulence
and mixing in the region and can dissipate energy away the tidal bands, which may
"decouple" the correlated response of $D_1$ and $D_2$.  This may also help explain the insignificant
value seen at Zhapo just to the west of Hong Kong within the influence of the river.  The
second sub-region is in the Taiwan Strait.  Here, there is a larger amount of semidiurnal
baroclinic energy than diurnal, as part of the $D_2$ wave that enters through the Luzon Strait
travels north through the Taiwan Strait to meet the incoming $D_2$ wave from the East China
Sea, leading to a pronounced resonance along the Taiwan coast (though not along the coast of

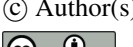



China, due to the irregular topography of the cross-section. (Jan et al., 2004).  However, there
is no significant diurnal wave or internal tide in the Taiwan Strait, so the semidiurnal
constituents dominate here, and is thus decoupled from the diurnal variability.

The OT TAC results in Hong Kong and the SCS show a generally negative relation

(Figure 6 (c) and (d)). The sensitivity of shallow-water overtides (OT) to fluctuations in the
forcing tides are most significant at harbor locations and along the southeastern reaches of
Hong Kong. In the SCS, OT TACs are most important in the Beibu Gulf and near the Taiwan
Strait, further suggesting the importance of friction in these dynamic regions. The strength of
forcing tides and shallow-water overtides should both be dependent on the depth and
morphology, so such a inverse relationship is to be expected in general. However, the
implications of the frictional response can be complex under scenarios of rising sea levels
(e.g., Hollemann and Stacey, 2014).  For a sea-level rise along a shore with a gently sloping
bottom, such as a beach, rising sea levels will inundate more low-lying areas, increasing
friction and dissipating energy from the forcing tides.  By contrast, harbors are deeper and
flat-walled, often deepened further to develop navigation channels or accommodate shipping
terminals.  For these regions, sea-level rise will decrease friction, as the distance from the
bottom will increase without new land areas being inundated, hence, less energy will be
dissipated from the forcing tides.  This in turn may have indirect effects on the total sea levels
in other regions near the deep harbor areas.  In either situation, the relations of OTs to forcing
tides will be negative.  Interestingly, the OT TACs are insignificant in the southern SCS;
since these regions are the shallowest in the study domain, they should be subject to large
frictional tides, yet they are not correlated to the forcing tides as they are in the northern SCS.
This may be at least partly attributable to the dominant importance of seasonal processes in
the Gulf of Thailand reported on by Devlin et al. (2018) and may be indicative of the OT
TAC relations in the northern SCS being more closely related to baroclinic activity than
water depth, since baroclinic energy is less important in very shallow regions.

*4.2 Effects of regional tidal variability on local variability*

The presence of strong $M_3$ and $MO_3$ tides at most gauges in Hong Kong (Fig 9)

indicates a connection to the dynamics at the shelf where significant $D_3$ energy has been
observed (Xie et al., 2008). The $N_2$ tide with its typical ~8.85 yr periodicity is largest in the
Taiwan Strait, but closer to Luzon and elsewhere in the SCS, $N_2$ is much smaller.  This
suggests that the source of the long-period signal in $M_3$ and $MO_3$ is the $N_2$ energy originating





in the Taiwan Strait. The $N_2$ wave may couple with the incoming $D_1$ and $D_2$ energy from
Luzon at the northern SCS shelf and may intensify PSI and triad interactions.  The $M_3$ and
$MO_3$ signals are likely initially generated near the shelf, and then may be enhanced by $N_2$
energy from the Taiwan Strait which imparts the long-period modulation to the $D_3$ band,
leading to coherent $D_3$ signals with long-period modulations observable in Hong Kong. A
resonance in $M_3$ has been observed before on the shelf near Brazil in the south Atlantic
(Huthnance, 1980), demonstrating that a large $M_3$ can result from a combination of an "organ
pipe" quarter wave resonance from the tide that leads to high amplitudes at the shore (Webb,
1976), and a half-wave transverse resonance that enhances the tides at the edge of the shelf.
Such a mechanism is also possible near Hong Kong, which is at a similar latitude as Brazil,
and the shelf in the SCS near Hong Kong has similar depth, width, and slope characteristics
as the Brazil shelf.   This hypothesis is further supported by noticing that the long-period
modulation is strongest in the Taiwan Strait and northern shelf region but diminishes further
away (Fig 10).  In the Beibu Gulf and the southern SCS, the $N_2$ variation is almost
nonexistent, and the $M_3$ signal is much smaller. Outside the SCS in the Pacific (Ishigaki), the
$M_3$ tide is virtually nonexistent, with no significant periodicity seen.

The usually insignificant $2N_2$ tide is also interesting, being more spatially coherent

than $N_2$ in Hong Kong after ~2000, before the anomalous event (Fig 9(b)).  This suggests that
the anomaly could be related to a resonance shift due to the combination of rising sea-levels
and the anthropogenically modified coastal morphology. Since the $N_2$ and $2N_2$ frequencies
are close (within 2%), is it possible that the extensive changes to the coastal morphology have
shifted the dominant resonance by a similar amount, yielding the anomaly event as a
harmonic adjustment to new forcing conditions. It may alternatively be related to a regional
change in the SCS (e.g., rising MSL or increased stratification due to upper-ocean warming).
However, since data coverage is sparse in the SCS, and few locations allow direct
comparisons of "before and after", any conclusions based on this limited data would be hasty.
Local and regional models may help to determine which spatial scale is most relevant.

Hong Kong has had a long history of land reclamation to accommodate an ever-

growing infrastructure and population, including the building of a new airport island (Chep
Lap Kok), land connections and from the Kowloon Peninsula to Stonecutters' Island and
channel deepening to accommodate container terminals, and many bridges, tunnels, and "new
cities", built on reclaimed land (e.g., Tai Po and Tseung Kwan O). All of these may have
changed the resonance and/or frictional properties of the region. Tai Po Kau has also seen



some land reclamation efforts, such as Science Park, that have changed the coastal morphology. Both locations also show coherent $D_1/D_2$ and OT TACs, as well as having the largest δ-HATs, and the largest tidal anomalies in the 2000s. Other locations in Hong Kong did not show such extreme variations, so these variations appear to be amplified in harbor areas. Decreases in friction associated with sea-level rise in the SCS may lead to higher forcing tides, and those changes may also be amplified by the close correlations of $D_1$ and $D_2$ variability or local harbor development which may further decrease local friction. Hence, a small change in friction due to a small sea-level change may induce a significant change in tidal amplitudes. The positive reinforcement of multiple tides correlated with regional sea-level adjustments may amplify the risks of coastal inundation and coastal flooding, as evidenced by the gauges that had the largest δ-HAT values.

*4.3 Limitations of this study and future steps*

The inventory of tide gauges provided by HKO and the HKMD has revealed new dynamics and spatial connectivity in the area. However, some gauges are of short length and/or riddled with data gaps, making a full analysis of the area problematic. For example, the Tsim Bei Tsui (TBT) gauge covers a long period, but there are significant gaps in the record, which complicates our analysis. This gauge is located within a harbor region (Deep Bay), bordered to the north by Shenzhen, PRC, which has also grown and developed its coastal infrastructure in past decades, therefore, one might expect similar dynamics are was seen at QB and TPK. While there were significant OT TACs, and $D_1/D_2$ correlations at TBT, no significant TACs or δ-HATs were observed. The large anomalies seen at QB and TPK around 2000 are suggested by the data at TBT, but some of the missing data corresponds to this time. Without more data or observations, no answers can be concluded about this location at the present time. However, future studies will examine this region via remote sensing and *in-situ* data to better understand the tidal behavior in this area, since the Deep Bay region is highly ecologically sensitive, being populated by extensive mangrove forests which may be disturbed by rapidly changing sea levels (Zhang et al., 2018), so accurate determination of future sea-levels is of utmost importance to the vitality of these important ecosystems.

Furthermore, there is only limited historical data available in the rest of the SCS, most of it having not been updated in 20 years. This complicates efforts to understand the full spatial and temporal extent of the tidal variability in the greater SCS region. A caveat is also





made about the very large TACs and δ-HATs observed in the Beibu Gulf; these are likely due
to the sensitive resonance in the Gulf, and it is unlikely that such large-magnitude changes
will remain linear over such large MSL fluctuations (i.e., it violates the "small-amplitude"
assumption taken above).  Yet, the behavior in the region is still worthy of future study.
Another limitation comes from the nature of the harmonic analysis technique used
(R_T_TIDE) which only resolves energy at discrete tidal frequencies.  This will not be able
to identify tidal energy at the local (latitude-dependent) inertial frequency, $f$ (at Hong Kong,
$T_f \sim 31.625$ hr), which may be a significant component of the energy cascade (Xie et al.,
2008; 2011;2013; Chinn et al., 2012).  It is also likely that the $M_1$ tide is part of the cascade,
yet this tide was below the noise limit at all gauges analyzed here.  However, since the $M_1$
interactions are an intermediate step that transfers energy to $M_3$ (i.e., $M_2$ to $M_1$, then to $M_3$ via
$M_2 + M_1$), this energy is high-frequency and not detectable at the yearly-analyzed scale.
Finally, there are only surface observations available (i.e., tide gauges), though the tidal
velocities are also variable at depth. The installation of current profilers at inland and
offshore locations near Hong Kong could provide beneficial observations of the three-
dimensional dynamics, could reveal the presence of energy at lesser frequencies such as $M_1$
and $f$ as well as being able to separate the baroclinic component of the tides. Previous current
profiler observations in the Hong Kong waters are currently being analyzed, to be presented
in a future study. Finally, the tidal variability could be better explored via utilization of
analytical and numerical models.  This is beyond the scope of the current observational study
but is the subject of an ongoing project.

### 5. Conclusions

This study has presented new information about the tidal variability in Hong Kong,

based on observations of a set of historical and modern tide gauges in Hong Kong and in the
South China Sea. The observed dynamics support the hypothesis that the changes are due to
multiple processes and are best described as an amplification of the local (Hong Kong) tidal
response to changes in the prevailing regional (SCS) tidal patterns, which may have been
enhanced by local harbor changes and land reclamation.  The $D_1/D_2$ and OT TACs, on the
other hand, are more likely due to the internal tide dynamics near the Luzon Strait which are
enhanced at the shelf; this may influence the tidal behavior in other parts of the SCS and may
also explain the large spatial scale of these correlations, as well as explaining the presence of
$M_3$.  The large TACs and δ-HATs in Hong Kong and the anomalous events in tidal
amplitudes seen at the Quarry Bay and Tai Po Kau gauges are likely due to a combination of





changing resonance and friction induced by coastal improvement projects which may amplify
the regional $D_1/D_2$ and OT TACs in harbor regions. These anomalies also suggest that a
regime change in tidal resonance has occurred, with the effect being most pronounced at
gauges in semi-enclosed harbors where all tidal components are strongly modulated via the
conservation of the $D_1/D_2$ ratios. A shift in the tidal regime is further suggested by the less
significant spatial correlations of most tidal components (except $2N_2$) observed in recent
years as compared to historical eras.
Overall, the tidal variability seen in Hong Kong may have significant impacts on the
future of total sea-levels in the region.  Short-term inundation events, such as nuisance
flooding, may be amplified under scenarios of higher sea-levels that lead to corresponding
changes in the tides, as evidenced by the strong $D_1/D_2$ and OT connections and very large
TACs which may amplify small changes in water levels or reductions in friction due to
harbor improvements.  It is probable that changes in harbor geometry have influenced tidal
evolution in Hong Kong as a cumulative effect of all projects.  Future studies will perform
simple analytical models as well as high resolution three-dimensional models to simulate
changing coastlines under a variety of sea-level, tidal forcing, and anthropogenic change
scenarios (historical and future) to better understand the tidal dynamics in Hong Kong at the
local scale (e.g., how much morphological change in a harbor region would be needed to shift
the dominant resonance from $N_2$ to $2N_2$), conditions that allow or enhance PSI or resonant
triad interactions, and the utilization satellite-derived tidal observations and models in the
South China Sea to better understand the dynamics at the regional scale, particularly the
$D_1/D_2$ ratios, and the $M_3$ prevalence in the SCS.










**Code availability** All code employed in this study was developed using MATLAB, version
R2011B. All code and methods can be provided upon request.

**Data Availability** The data used in this study from the Hong Kong Observatory (HKO;
www.hko.gov.hk) and the Hong Kong Marine Department (HKMD;
www.mardep.gov.hk/en/home.html) was provided upon request, discussion of intentions of
use, and permission from the appropriate agency supervisors. Data used from the University
of Hawaii Sea Level Center (UHSLC; www.uhslc.soest.hawaii.edu) is publicly available.

**Author Contributions** ATD did all analyses, figures, tables, the majority of writing, and
complied the manuscript. JP provided editing, insight, guidance, and direction to this study.
HL provided critical and helpful input.

**Competing Interests** The authors declare they have no competing interest.

**Acknowledgements** This work is supported by The National Basic Research Program of
China (2015CB954103), the National Natural Science Foundation of China (project
41376035), the General Research Fund of Hong Kong Research Grants Council (RGC)
(CUHK 402912 and 403113), the Hong Kong Innovation and Technology Fund under the
grants (ITS/259/12 and ITS/321/13), and the direct grants of the Chinese University of Hong
Kong.



































**FIGURE CAPTIONS:**

**Figure 1** Tide gauge locations in Hong Kong used in this study. Green markers indicate active gauges provided by the Hong Kong Observatory (HKO), light blue markers indicate gauges provided by the Hong Kong Marine Department (HKMD), and red markers indicate historical gauges once maintained by HKO that are no longer operational.

**Figure 2** Tide gauge locations in the South China Sea (SCS). All tide gauge data is provided by the University of Hawaii Sea Level Center; green markers indicate actively recording and updated tide gauges, and red markers indicate historical gauges that have not been publicly updated since 1997.

**Figure 3** Semidiurnal tidal anomaly correlations (TACs) of detrended $M_2$ amplitude to detrended MSL in (a) Hong Kong, (b) the South China Sea, and of detrended $S_2$ amplitude to detrended MSL in (c) Hong Kong, and (d) the South China Sea. Red markers indicate positive TACs and blue indicates negative TACs, with the marker size showing the relative magnitude according to the legend. Black marks indicate insignificant TACs. Map backgrounds in (b) and (d) show mean tidal amplitudes over the period of 1993-2014 (color scale, meters) and phases (solid lines, 30° increment), taken from the ocean tidal model of TPXO7.2, (Egbert and Erofeeva, 2002, 2010).

**Figure 4** Diurnal tidal anomaly correlations (TACs) of detrended $K_1$ amplitude to detrended MSL in (a) Hong Kong, (b) the South China Sea, and of detrended $O_1$ amplitude to detrended MSL in (c) Hong Kong, and (d) the South China Sea. Red markers indicate positive TACs and blue indicates negative TACs, with the marker size showing the relative magnitude according to the legend. Black marks indicate insignificant TACs. Map backgrounds in (b) and (d) show mean tidal amplitudes over the period of 1993-2014 (color scale, meters) and phases (solid lines, 30° increment), taken from the ocean tidal model of TPXO7.2, (Egbert and Erofeeva, 2002, 2010).

**Figure 5** Results of the $\delta$-HAT$_4$ determinations, the correlation of detrended ($M_2 + S_2 + K_1 + O_1$) to detrended MSL in Hong Kong (a) and the SCS (b), and results of the $\delta$-HAT$_8$ determinations, the correlation of detrended ($M_2 + S_2 + N_2 + K_2 + K_1 + O_1 + P_1 + Q_1$) to detrended MSL in Hong Kong (c) and the SCS (d). Red markers indicate positive TACs and blue indicates negative TACs, with the marker size showing the relative magnitude according to the legend. Black marks indicate insignificant TACs.

**Figure 6** Results of the $D_1/D_2$ TACs, the correlation of detrended $D_2$ ($M_2 + S_2 + N_2 + K_2$) to detrended $D_1$ ($K_1 + O_1 + P_1 + Q_1$) in Hong Kong (a) and the SCS (b), and results of the OT TACs, the correlation of detrended ($D_1 + D_2$) to detrended OT($M_4 + M_6 + MK_3 + MO_3 + MS_4 + MN_4 + S_4$) in Hong Kong (c) and the SCS (d). Red markers indicate positive TACs and blue indicates negative TACs, with the marker size showing the relative magnitude according to the legend. Black marks indicate insignificant TACs.

**Figure 7** Time series of water level spectrum components at the Quarry Bay (QB; blue) and Tai Po Kau (TPK; red) tide gauges in Hong Kong, showing the $D_1$ band (a), the $D_2$ band (b), the OT band (c) and mean sea-level (MSL) (d). Components are plotted as a function of normalized amplitudes to show relative variability, with mean values given in the legend.



**Figure 8** Time-series of the detrended $D_2$ water level spectrum component at all tide gauges
in Hong Kong, plotted as a normalized amplitude to show relative variability, with mean
values given in the legend.  Each gauge is indicated by color according to the legend, with the
QB (solid blue) and TPK (solid red) gauges shown as heavier lines.  Horizontal dotted lines
indicate the $\pm 5\%$ variational band relative to the mean amplitude.
**Figure 9** Minor constituent variability at selected Hong Kong gauges.  $N_2$ is shown in (a),
$2N_2$ in (b), $M_3$ in (c) and $MO_3$ in (d).  All quantities are plotted as normalized amplitudes to
show relative variability, with mean values given in the legends at the right.
**Figure 10** Minor constituent variability at selected South China Sea gauges.  $N_2$ is shown in
(a), $2N_2$ in (b), $M_3$ in (c) and $MO_3$ in (d).  All quantities are plotted as normalized amplitudes
to show relative variability, with mean values given in the legends at the right.
























**FIGURES**

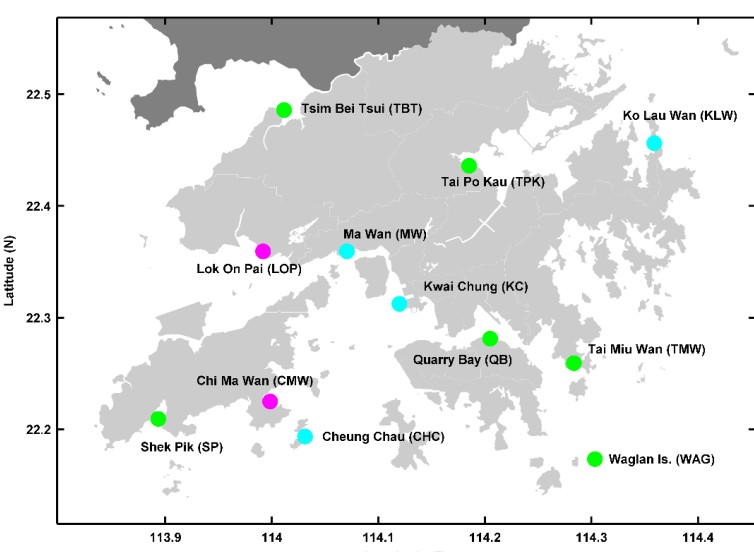


**Figure 1** Tide gauge locations in Hong Kong used in this study.  Green markers indicate
active gauges provided by the Hong Kong Observatory (HKO), light blue markers indicate
gauges provided by the Hong Kong Marine Department (HKMD), and red markers indicate
historical gauges once maintained by HKO that are no longer operational.

















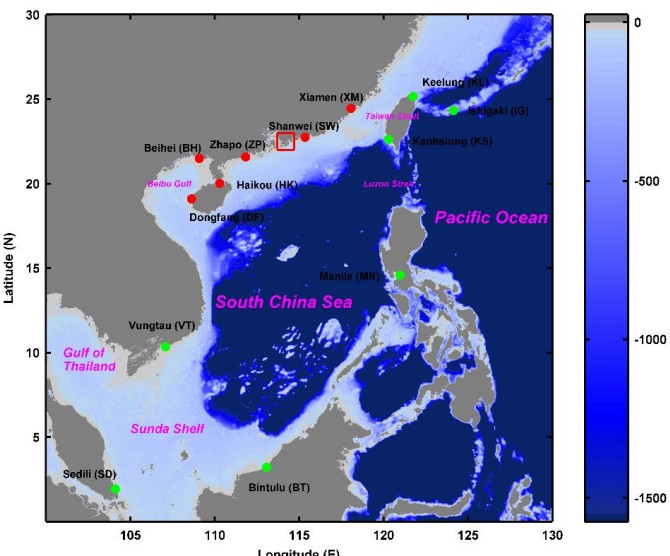


**Figure 2** Tide gauge locations in the South China Sea (SCS). All tide gauge data is provided
by the University of Hawaii Sea Level Center; green markers indicate actively recording and
updated tide gauges, and red markers indicate historical gauges that have not been publicly
updated since 1997.















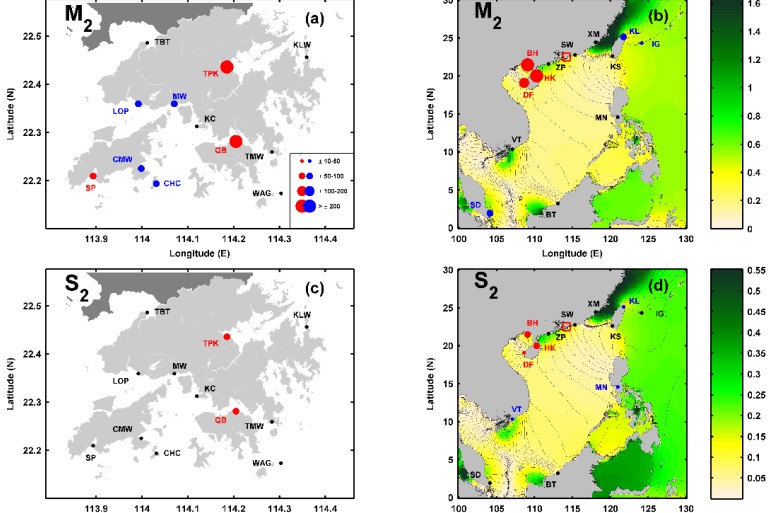


**Figure 3** Semidiurnal tidal anomaly correlations (TACs) of detrended $M_2$ amplitude to
detrended MSL in (a) Hong Kong, (b) the South China Sea, and of detrended $S_2$ amplitude to
detrended MSL in (c) Hong Kong, and (d) the South China Sea.  Red markers indicate
positive TACs and blue indicates negative TACs, with the marker size showing the relative
magnitude according to the legend. Black marks indicate insignificant TACs.  Map
backgrounds in (b) and (d) show mean tidal amplitudes over the period of 1993-2014 (color
scale, meters) and phases (solid lines, 30° increment), taken from the ocean tidal model of
TPXO7.2, (Egbert and Erofeeva, 2002, 2010).
















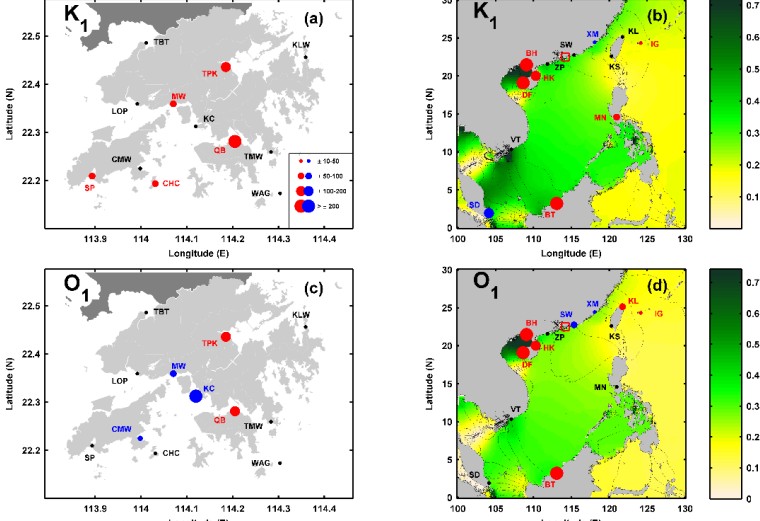


**Figure 4** Diurnal tidal anomaly correlations (TACs) of detrended $K_1$ amplitude to detrended MSL in (a) Hong Kong, (b) the South China Sea, and of detrended $O_1$ amplitude to detrended MSL in (c) Hong Kong, and (d) the South China Sea. Red markers indicate positive TACs and blue indicates negative TACs, with the marker size showing the relative magnitude according to the legend. Black marks indicate insignificant TACs. Map backgrounds in (b) and (d) show mean tidal amplitudes over the period of 1993-2014 (color scale, meters) and phases (solid lines, 30° increment), taken from the ocean tidal model of TPXO7.2, (Egbert and Erofeeva, 2002, 2010).
















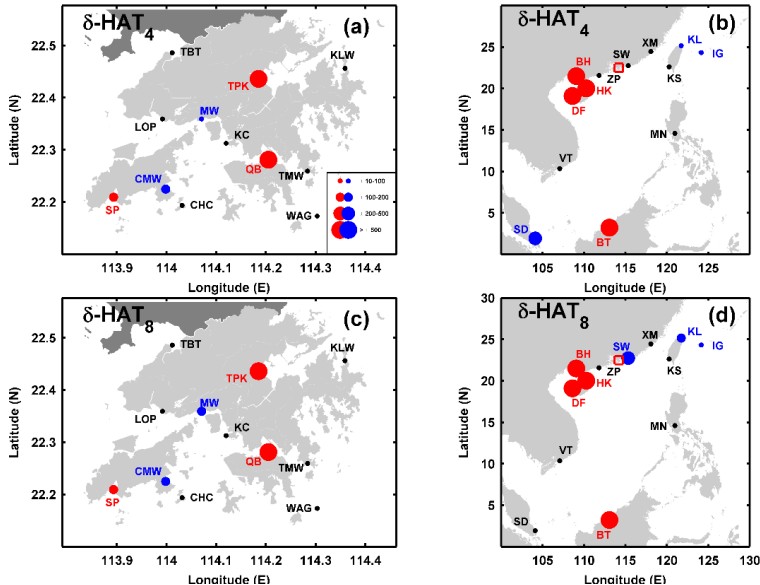


**Figure 5** Results of the δ-HAT$_4$ determinations, the correlation of detrended ($M_2 + S_2 + K_1 + O_1$) to detrended MSL in Hong Kong (a) and the SCS (b), and results of the δ-HAT$_8$ determinations, the correlation of detrended ($M_2 + S_2 + N_2 + K_2 + K_1 + O_1 + P_1 + Q_1$) to detrended MSL in Hong Kong (c) and the SCS (d). Red markers indicate positive TACs and blue indicates negative TACs, with the marker size showing the relative magnitude according to the legend. Black marks indicate insignificant TACs.













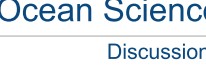
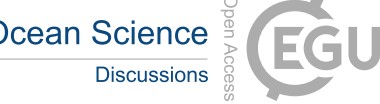


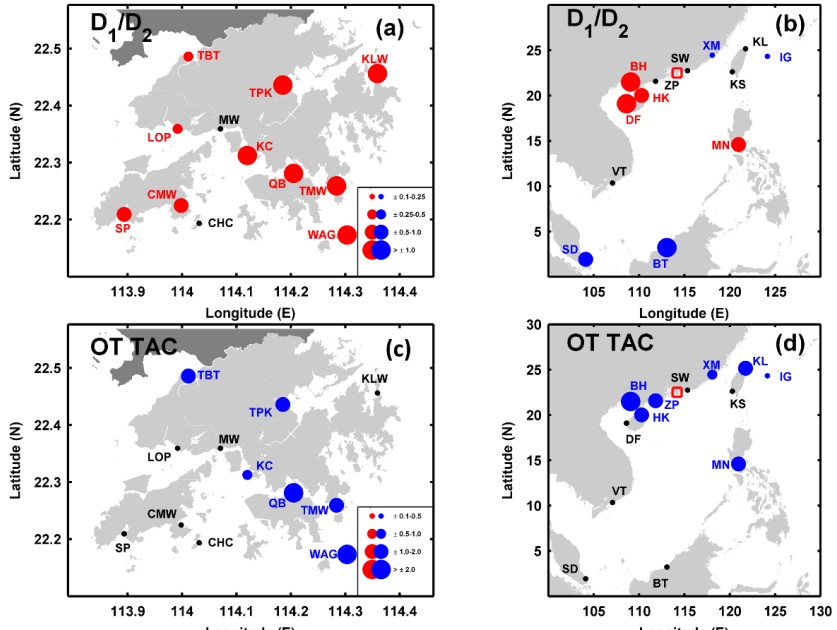


**Figure 6** Results of the $D_1/D_2$ TACs, the correlation of detrended $D_2$ ($M_2 + S_2 + N_2 + K_2$) to
detrended $D_1$ ($K_1 + O_1 + P_1 + Q_1$) in Hong Kong (a) and the SCS (b), and results of the OT
TACs, the correlation of detrended ($D_1 + D_2$) to detrended OT($M_4 + M_6 + MK_3 + MO_3 + MS_4$
$+ MN_4 + S_4$) in Hong Kong (c) and the SCS (d). Red markers indicate positive TACs and
blue indicates negative TACs, with the marker size showing the relative magnitude according
to the legend. Black marks indicate insignificant TACs.
















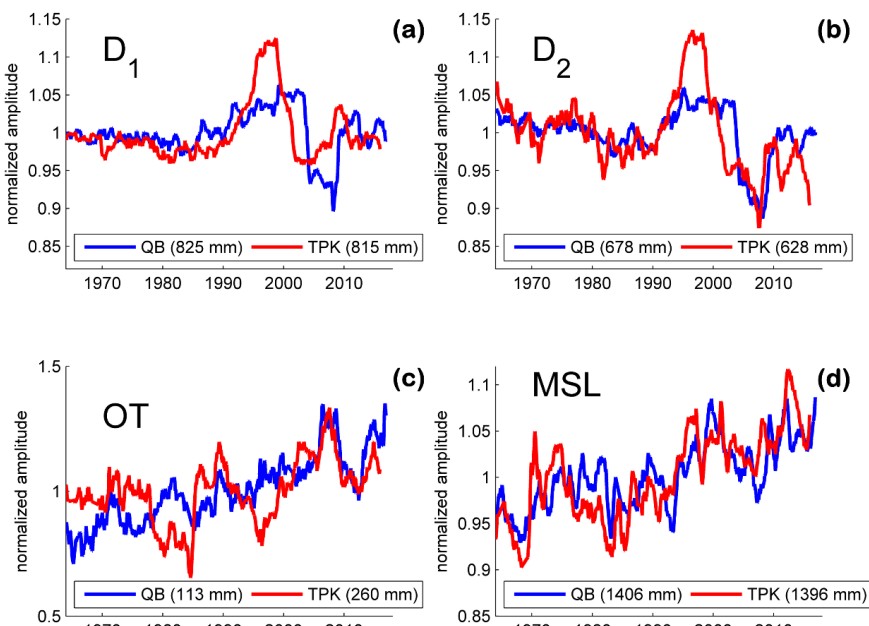


**Figure 7** Time series of water level spectrum components at the Quarry Bay (QB; blue) and
Tai Po Kau (TPK; red) tide gauges in Hong Kong, showing the $D_1$ band (a), the $D_2$ band (b),
the OT band (c) and mean sea-level (MSL) (d). Components are plotted as a function of
normalized amplitudes to show relative variability, with mean values given in the legend.















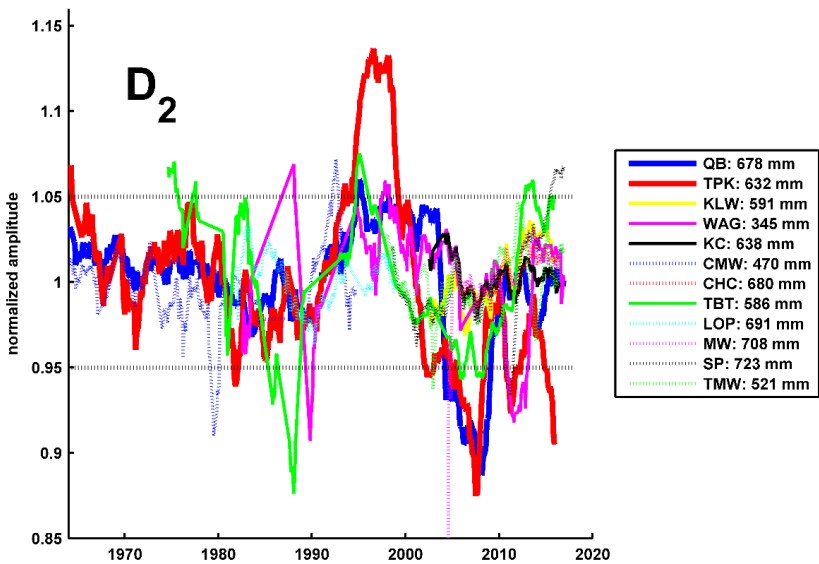


**Figure 8** Time-series of detrended $D_2$ at all tide gauges in Hong Kong, plotted as a
normalized amplitude to show relative variability, with mean values given in the legend.
Each gauge is indicated by color according to the legend, with the QB (solid blue) and TPK
(solid red) gauges shown as heavier lines. Horizontal dotted lines indicate the ±5%
variational band relative to the mean amplitude.















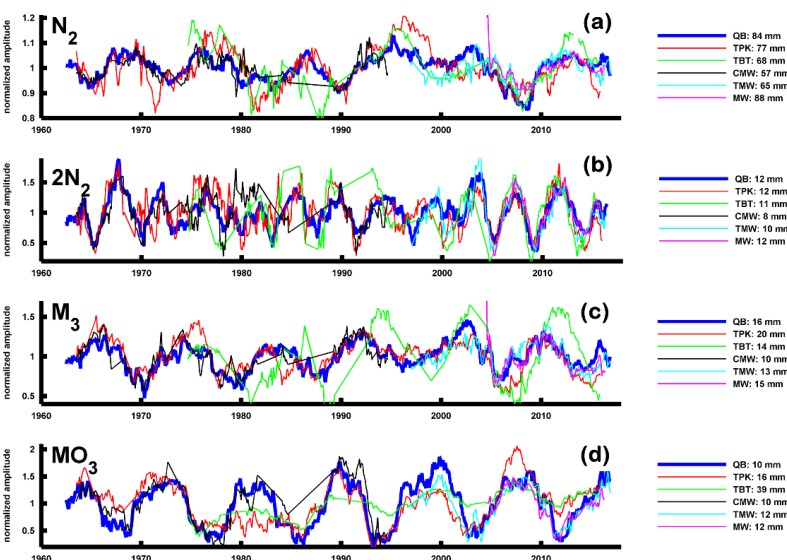


**Figure 9** Minor constituent variability at selected Hong Kong gauges. $N_2$ is shown in (a), $2N_2$ in (b), $M_3$ in (c) and $MO_3$ in (d). All quantities are plotted as normalized amplitudes to show relative variability, with mean values given in the legends at the right.


















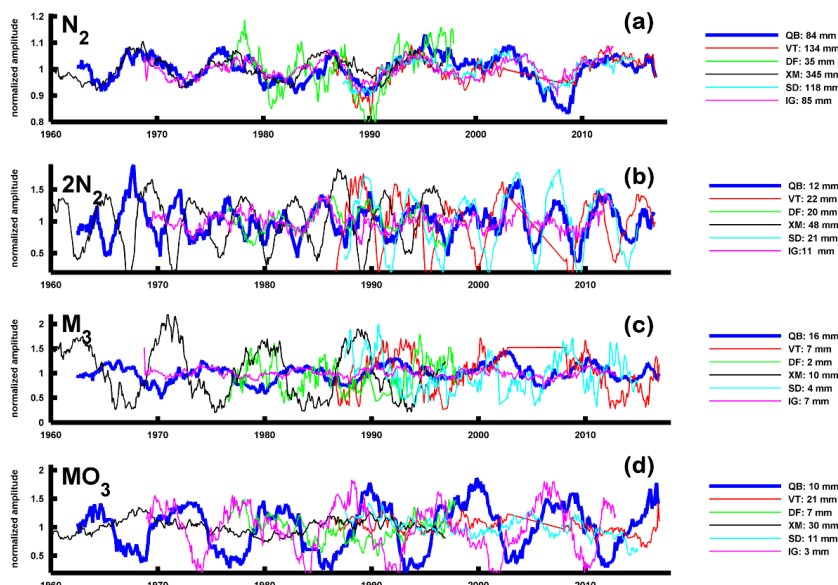


**Figure 10** Minor constituent variability at selected South China Sea gauges. $N_2$ is shown in (a), $2N_2$ in (b), $M_3$ in (c) and $MO_3$ in (d). All quantities are plotted as normalized amplitudes to show relative variability, with mean values given in the legends at the right.




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

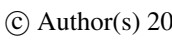



**TABLES:**
**Table 1** Metadata for all tide gauge locations, giving latitude/longitude, and star year/end
year of data analyzed.  Except where indicated by country code, all locations are located in
the People's Republic of China (PRC). The solid horizontal line demarcates Hong Kong and
South China Sea tide gauges.

| Station | Latitude | Longitude | Start Year | End Year |
|---|---|---|---|---|
| Quarry Bay (QB) | 22.27° N | 114.21° E | 1954 | 2016 |
| Tai Po Kau (TPK) | 22.42° N | 114.19° E | 1963 | 2016 |
| Tsim Bei Tusi (TBT) | 22.48° N | 114.02° E | 1974 | 2016 |
| Chi Ma Wan (CMW) | 22.22° N | 114.00° E | 1963 | 1997 |
| Cheung Chau (CHC) | 22.19° N | 114.03° E | 2004 | 2016 |
| Lok On Pai (LOP) | 22.35° N | 114.00° E | 1981 | 1999 |
| Ma Wan (MW) | 22.35° N | 114.06° E | 2004 | 2016 |
| Tai Miu Wan (TMW) | 22.26° N | 114.29° E | 1996 | 2016 |
| Shek Pik (SP) | 22.21° N | 113.89° E | 1999 | 2016 |
| Waglan Island (WAG) | 22.17° N | 114.30° E | 1995 | 2016 |
| Ko Lau Wan (KLW) | 22.45° N | 114.34° E | 2004 | 2016 |
| Kwai Chung (KC) | 22.31° N | 114.12° E | 2004 | 2016 |
| Dongfang (DF) | 19.10° N | 108.62° E | 1975 | 1997 |
| Beihei (BH) | 21.48° N | 109.08° E | 1975 | 1997 |
| Haikou (HK) | 20.02° N | 110.28° E | 1976 | 1997 |
| Zhapo (ZP) | 21.58° N | 111.83° E | 1975 | 1997 |
| Shanwei (SW) | 22.75° N | 115.35° E | 1975 | 1997 |
| Xiamen (XM) | 24.45° N | 118.07° E | 1954 | 1997 |
| Keelung (KL) | 22.62° N | 120.29° E | 1980 | 2014 |
| Kaohsiung (KS) | 25.16° N | 121.75° E | 1980 | 2014 |
| Manila, PHL (MN) | 14.59° N | 120.97° E | 1984 | 2016 |
| Vung Tau, VTM (VT) | 10.34° N | 107.07° E | 1986 | 2014[*] |
| Sedili, MLY (SD) | 1.93° N | 104.12° E | 1986 | 2016 |
| Bintulu, MLY (BT) | 3.22° N | 113.07° E | 1992 | 2016 |
| Ishigaki, JPN (IG) | 24.33° N | 124.15° E | 1968 | 2013 |

[*]-missing data from 2002-2007














**Table 2** Amplitude TACs for $M_2$, $S_2$, $K_1$, and $O_1$. All values given are in units of milimeter
change in tidal amplitude for a 1-meter fluctuation in sea-level (mm m$^{-1}$).  Statistically
significant positive values are given in bold italic text.

| Station | $M_2$ TAC | $S_2$ TAC | $K_1$ TAC | $O_1$ TAC |
|---|---|---|---|---|
| *Quarry Bay (QB)* | ***+218 ± 37*** | ***+85 ± 16*** | ***+220 ± 15*** | ***+146 ± 11*** |
| *Tai Po Kau (TPK)* | ***+267 ± 42*** | ***+98 ± 17*** | ***+190 ± 68*** | ***+100 ± 25*** |
| *Tsim Bei Tusi (TBT)* | +7 ± 80 | -10 ± 15 | +32 ± 22 | +24 ± 22 |
| *Chi Ma Wan (CMW)* | ***-58 ± 11*** | -7 ± 5 | -18 ± 8 | ***-37 ± 10*** |
| *Cheung Chau (CHC)* | ***-63 ± 20*** | -22 ± 35 | ***+69 ± 48*** | +50 ± 92 |
| *Lok On Pai (LOP)* | ***-81 ± 24*** | -18 ± 8 | +8 ± 32 | -24 ± 12 |
| *Ma Wan (MW)* | ***-68 ± 4*** | +1 ± 25 | ***+52 ± 4*** | ***-62 ± 21*** |
| *Tai Miu Wan (TMW)* | +22 ± 59 | -1 ± 9 | +10 ± 22 | +3 ± 8 |
| *Shek Pik (SP)* | ***+62 ± 29*** | +11 ± 18 | ***+70 ± 4*** | +28 ± 17 |
| *Waglan Island (WAG)* | +1 ± 21 | +3 ± 6 | +9 ± 7 | -9 ± 8 |
| *Ko Lau Wan (KLW)* | -46 ± 39 | -11 ± 17 | +29 ± 65 | +60 ± 57 |
| *Kwai Chung (KC)* | -90 ± 46 | -10 ± 29 | -91 ± 226 | ***-202 ± 161*** |
| *Dongfang (DF)* | ***+190 ± 75*** | ***+43 ± 9*** | ***+482 ± 53*** | ***+320 ± 52*** |
| *Beihei (BH)* | ***+461 ± 170*** | ***+88 ± 19*** | ***+579 ± 152*** | ***+294 ± 78*** |
| *Haikou (HK)* | ***+379 ± 106*** | ***+55 ± 8*** | ***+180 ± 28*** | ***+194 ± 37*** |
| *Zhapo (ZP)* | -32 ± 30 | -12 ± 30 | +40 ± 33 | +1 ± 44 |
| *Shanwei (SW)* | +30 ± 30 | -34 ± 31 | -26 ± 15 | ***-79 ± 53*** |
| *Xiamen (XM)* | +93 ± 31 | -32 ± 35 | ***-46 ± 4*** | ***-48 ± 8*** |
| *Keelung (KL)* | ***-69 ± 14*** | ***-37 ± 5*** | -4 ± 8 | ***+21 ± 4*** |
| *Kaohsiung (KS)* | +25 ± 8 | +1 ± 18 | +1 ± 8 | +28 ± 16 |
| *Manila, PHL (MN)* | -17 ± 16 | ***-21 ± 9*** | ***+83 ± 12*** | -20 ± 16 |
| *Vung Tau, VTM (VT)* | +21 ± 26 | ***-44 ± 7*** | +7 ± 21 | +20 ± 6 |
| *Sedili, MLY (SD)* | ***-72 ± 35*** | +24 ± 24 | ***-148 ± 35*** | -54 ± 33 |
| *Bintulu, MLY (BT)* | -37 ± 15 | +11 ± 7 | ***+291 ± 45*** | ***+320 ± 36*** |
| *Ishigaki, JPN (IG)* | ***-46 ± 2*** | -8 ± 7 | ***+23 ± 11*** | +1 ± 11 |



















**Table 3** The $\delta$-HAT$_4$, $\delta$-HAT$_8$, D$_1$/D$_2$ TACs, and OT TACs. The $\delta$-HAT values given are in units of milimeter change in tidal amplitude for a 1-meter fluctuation in sea-level (mm m$^{-1}$). D$_1$/D$_2$ and OT TACs are in unitless ratios (i.e., mm mm$^{-1}$) Statistically significant positive values are given in bold italic text.

| Station | $\delta$-HAT$_4$ | $\delta$-HAT$_8$ | D$_1$/D$_2$ | OT/ (D$_1$ + D$_2$) |
|---|---|---|---|---|
| *Quarry Bay (QB)* | *+665 ± 82* | *+834 ± 108* | *+1.08 ± 0.05* | *-3.62 ± 0.99* |
| *Tai Po Kau (TPK)* | *+612 ± 210* | *+797 ± 138* | *+1.01 ± 0.04* | *-1.87 ± 0.10* |
| *Tsim Bei Tusi (TBT)* | +56 ± 117 | +41 ± 180 | *+0.37 ± 0.02* | *-1.69 ± 0.14* |
| *Chi Ma Wan (CMW)* | *-119 ± 19* | *-159 ± 28* | *+0.74 ± 0.19* | -0.01 ± 0.60 |
| *Cheung Chau (CHC)* | -12 ± 42 | +224 ± 646 | +0.81 ± 1.03 | -0.11 ± 1.36 |
| *Lok On Pai (LOP)* | -114 ± 45 | -112 ± 110 | *+0.26 ± 0.05* | -0.26 ± 0.21 |
| *Ma Wan (MW)* | *-91 ± 73* | *-117 ± 35* | +0.57 ± 1.02 | -0.42 ± 1.44 |
| *Tai Miu Wan (TMW)* | +42 ± 100 | +89 ± 99 | *+1.04 ± 0.20* | *-1.31 ± 0.23* |
| *Shek Pik (SP)* | *+138 ± 37* | *+183 ± 20* | *+0.89 ± 0.06* | -0.01 ± 0.60 |
| *Waglan Island (WAG)* | +3 ± 31 | +4 ± 30 | *+1.11 ± 0.17* | *-3.05 ± 0.43* |
| *Ko Lau Wan (KLW)* | -66 ± 47 | +83 ± 367 | *+1.31 ± 0.62* | -0.35 ± 0.82 |
| *Kwai Chung (KC)* | -55 ± 64 | +270 ± 730 | *+1.19 ± 0.60* | *-0.62 ± 0.42* |
| *Dongfang (DF)* | *+1037 ± 453* | *+1236 ± 113* | *+2.86 ± 0.19* | *-6.10 ± 2.69* |
| *Beihei (BH)* | *+1405 ± 453* | *+2190 ± 151* | *+1.22 ± 0.03* | *-5.21 ± 0.15* |
| *Haikou (HK)* | *+813 ± 217* | *+1086 ± 189* | *+0.61 ± 0.05* | *-1.75 ± 0.04* |
| *Zhapo (ZP)* | -34 ± 111 | -16 ± 69 | +0.14 ± 0.07 | *-1.69 ± 0.57* |
| *Shanwei (SW)* | -94 ± 94 | *-217 ± 150* | +0.02 ± 0.18 | -0.09 ± 0.20 |
| *Xiamen (XM)* | +54 ± 38 | -3 ± 43 | *+0.12 ± 0.04* | *-0.92 ± 0.23* |
| *Keelung (KL)* | *-95 ± 21* | *-125 ± 44* | +0.08 ± 0.11 | *-1.29 ± 0.57* |
| *Kaohsiung (KS)* | +54 ± 36 | +52 ± 83 | +0.16 ± 0.07 | -1.55 ± 0.74 |
| *Manila, PHL (MN)* | +39 ± 67 | +5 ± 53 | *+0.81 ± 0.61* | *-1.86 ± 0.49* |
| *Vung Tau, VTM (VT)* | -28 ± 22 | -11 ± 59 | +0.15 ± 0.08 | +0.40 ± 0.59 |
| *Sedili, MLY (SD)* | *-254 ± 70* | -76 ± 55 | *-0.63 ± 0.06* | *-1.33 ± 0.50* |
| *Bintulu, MLY (BT)* | *+600 ± 52* | *+942 ± 55* | *-3.81 ± 1.60* | +1.62 ± 0.98 |
| *Ishigaki, JPN (IG)* | *-58 ± 6* | +4 ± 24 | *-0.12 ± 0.09* | +0.31 ± 0.61 |


















**Table 4** Correlations of tidal components with the North Point/Quarry Bay (QB) tide gauge,
showing $M_2$, $K_1$, $N_2$, $2N_2$, $M_3$, and $MO_3$. Two numbers are given in each column,
representing the correlations in the "historical" era (pre-1997), and the "modern" era (post-
1997). Non-existent data is indicated by "~". An average value is also calculated at the local
(Hong Kong) and regional (South China Sea) scale for each era. Data records that cover both
time periods will indicate the better correlated era by bold text. Other tidal component
correlations (including MSL) are given in Table S3 in the supplementary material.

| Station | $M_2$ | $K_1$ | $N_2$ | $2N_2$ | $M_3$ | $MO_3$ |
|---|---|---|---|---|---|---|
| TPK | **0.83**/0.56 | **0.72**/0.30 | **0.71**/0.57 | 0.54/**0.73** | 0.76/**0.77** | 0.74/**0.78** |
| TBT | 0.58/**0.77** | **0.48**/0.19 | 0.72/**0.78** | 0.48/**0.70** | 0.45/**0.52** | 0.66/**0.78** |
| CMW/CHC | 0.49/**0.56** | **0.42**/0.21 | **0.69**/0.61 | 0.61/**0.94** | **0.88**/0.80 | **0.92**/0.90 |
| LOP/MW | **0.57**/0.11 | **0.55**/0.16 | **0.87**/0.76 | 0.74/**0.95** | **0.85**/0.29 | **0.88**/0.87 |
| TMW | ~/0.25 | ~/0.60 | ~/0.65 | ~/0.87 | ~/0.76 | ~/0.93 |
| SP | ~/0.30 | ~/0.56 | ~/0.59 | ~/0.83 | ~/0.59 | ~/0.83 |
| WAG | ~/0.22 | ~/0.52 | ~/0.62 | ~/0.82 | ~/0.76 | ~/0.90 |
| KC | ~/0.20 | ~/0.25 | ~/0.76 | ~/0.93 | ~/0.82 | ~/0.92 |
| KLW | ~/0.16 | ~/-0.02 | ~/0.70 | ~/0.92 | ~/0.76 | ~/0.94 |
| HK Ave. | **0.62**/0.28 | **0.54**/0.31 | **0.75**/0.67 | 0.59/**0.86** | **0.74**/0.67 | 0.80/**0.88** |
| DF | 0.78/~ | 0.62/~ | 0.63/~ | 0.63/~ | -0.32/~ | -0.27/~ |
| BH | 0.75/~ | 0.58/~ | 0.55/~ | 0.35/~ | -0.03/~ | 0.13/~ |
| HK | 0.82/~ | 0.53/~ | 0.61/~ | 0.27/~ | 0.18/~ | 0.21/~ |
| ZP | 0.34/~ | 0.68/~ | 0.78/~ | 0.12/~ | 0.75/~ | 0.64/~ |
| SW | 0.73/~ | 0.32/~ | 0.83/~ | 0.77/~ | 0.84/~ | 0.89/~ |
| XM | -0.49/~ | 0.24/~ | 0.61/~ | -0.47/~ | -0.63/~ | -0.15/~ |
| KL | ~/-0.32 | ~/-0.13 | ~/0.49 | ~/0.18 | ~/0.45 | ~/-0.37 |
| KS | ~/0.34 | ~/0.62 | ~/0.53 | ~/0.53 | ~/0.14 | ~/-0.10 |
| MN | ~/-0.16 | ~/-0.07 | ~/0.06 | ~/0.50 | ~/0.48 | ~/0.35 |
| VT | ~/0.49 | ~/0.63 | ~/0.56 | ~/0.08 | ~/0.54 | ~/-0.03 |
| SD | ~/0.40 | ~/-0.46 | ~/0.80 | ~/0.79 | ~/-0.03 | ~/-0.31 |
| BT | ~/0.10 | ~/0.54 | ~/0.19 | ~/0.21 | ~/0.19 | ~/-0.17 |
| IG | 0.18/**0.36** | **0.38**/0.07 | 0.62/**0.72** | 0.34/**0.54** | **0.52**/0.47 | -0.17/**0.08** |
| SCS Ave. | **0.45**/0.17 | **0.48**/0.17 | **0.66**/0.48 | 0.29/**0.41** | 0.19/**0.32** | **0.12**/-0.09 |


