# Peer review of "Tidal variability in the Hong Kong region"

_Ocean Science, 2018_

## Short Comment (SC1) · 11 Oct 2018

Richard D. Ray

*NASA Goddard Space Flight Center, Greenbelt, Maryland, USA*

Changes in oceanic tides—over periods from secular to intra-annual—have been discovered in many ports and are the subject of numerous investigations. Searching for correlations between changes in tidal coefficients and various other environmental variables such as mean sea level is an obvious first step toward understanding the mechanisms behind these changes. Such correlation exercises, however, require that the underlying tidal analysis is carefully done and that the resulting tidal time series (say, of tidal amplitudes) is a legitimate expression of variability in the ocean. That is, care must be taken to ensure that the time series is truly reflecting a change in the tide and is not merely an artifact of tidal analysis. If a tidal time series has evidence of periodicities at 18.6, 8.8, or 4.4 years, which are all well-understood modulations existing in the tidal potential, then it is surely a strong warning that artifacts are present. Computing correlations with such corrupted time series would not be very useful.

In the submitted paper by Devlin et al. (2018) one finds several suspect time series with evident oscillations; see their Figures 9 and 10. My purpose in writing is to describe how these oscillations arise. Ultimately they stem from inadequacies in the tidal analysis, which are sometimes not easily recognized or understood.

The constituents under discussion are $N_2$, $2N_2$, $M_3$, and $MO_3$. The first two are standard semidiurnal constituents arising from the primary degree-2 terms of the tidal potential. The third is a terdiurnal constituent arising from a degree-3 term. The last is a nonlinear compound tide arising from interactions between the linear constituents $M_2$ and $O_1$. Devlin et al. (2018) estimate amplitudes and phases of these constituents every year from a half-century of hourly data and show the resulting amplitudes in their Figures 9 and 10. All show clear periodicities or modulations. These arise from tidal spectral lines that are inseparable in one-year analyses.

Here I examine only the time series from Hong Kong, comprising two segments from tide gauges at North Point (1962–1985) and Quarry Bay (1986–2016), locations separated by approximately 1.7 km. Tidal analysis of the entire (24 years) North Point hourly time series yields, for the frequency bands of interest

Table 1: Selected tidal constants for North Point, Hong Kong.

| Cluster | Constituent | Doodson number | Amplitude (mm) | Phase (deg) |
|---------|-------------|----------------|----------------|-------------|
| $2N_2$ | $2MK_2$ | 235.555 | 3.2 | 294° |
| | $2N_2'$ | 235.655 | 1.3 | 231° |
| | $2N_2$ | 235.755 | 12.0 | 339° |
| $N_2$ | $N_2'$ | 245.555 | 0.9 | 255° |
| | $N_2$ | 245.655 | 83.2 | 356° |
| $MO_3$ | $MO_3$ | 345.555 | 7.8 | 325° |
| | $F_3$ | 345.655 | 4.2 | 281° |
| $M_3$ | $M_3$ | 355.555 | 14.3 | 307° |
| | $NK_3$ | 355.655 | 2.5 | 353° |

here, harmonic constants listed in Table 1. The table is split into four clusters of tidal spectral lines. The lines within each cluster form, nominally, a tidal constituent, since the first three digits of their Doodson numbers are identical. Their frequencies are thus all separated by less than 1 cpy. In an analysis of a single year of data, these lines are inseparable. Thus, a time series of annual constituent estimates must yield a modulated time series of the form displayed by Devlin et al. I have reproduced the results of Devlin et al. by solving for annual estimates from the combined North Point and Quarry Bay hourly time series; these are shown as solid circles in Figure 1. The estimates agree fairly well with those given by Devlin et al., except I find a smaller modulation in $N_2$.

It is interesting to understand the source of some of the tidal lines listed in Table 1. In both $N_2$ and $2N_2$ there are lines that arise from the third-degree terms of the tidal potential (Cartwright & Tayler, 1971, p. 69), which I denote with a prime symbol. Their Doodson numbers differ from those of the standard degree-2 constituents by one unit in the fourth digit. Since the fourth digit corresponds to the coefficient of the mean longitude of the moon's perigee (period 8.8 years), these degree-2 and degree-3 lines differ in frequency by 1 cycle in 8.8 years. Nominally one therefore needs at least 9 years of observations to separate the degree-2 and degree-3 constituents. In addition, $2N_2$ is perturbed by the compound tide $2MK_2$, thus bringing in a 4.4-year modulation.

[On a technical note, it is important to realize that forming tidal admittances,

[Figure]

Figure 1: Four tidal constituents at Hong Kong (North Point and Quarry Bay). Solid circles: annual amplitude estimates based on analyses of hourly data, with error bars based on the spectral energy of the tidal residuals integrated over a small window surrounding each constituent. Solid lines: predicted amplitude modulations based on the harmonic constants of Table 1, which accounts for the existance of clustered tidal spectral lines that are inseparable in yearly analyses.

as was done by Devlin et al., cannot automatically account for the presence of both degree-2 and degree-3 tides, even though this approach is indeed useful for accounting for some close spectral lines (e.g, it can usually account for the lines associated with 18.6-y nodal modulations). Admittances must be computed separately for degree-2 and degree-3 tides. Although their frequencies are nearly identical, their spatial forcing patterns over the global ocean are markedly different, so the ocean's response to these different components is also very different. See, for example, the approach to admittance calculations taken by Munk & Cartwright (1966) and the very different ocean responses for degree-2 and degree-3 diurnal tides observed by Cartwright (1975).]

The amplitude of $N_2'$ is small relative to the dominant $N_2$ line, so one should expect only small modulations in annual estimates of $N_2$. In other locations, the degree-3 constituent is larger and the resulting modulations more pronounced. One such example occurs in the Gulf of Maine (Doodson, 1924).

In the terdiurnal band, the linear $M_3$ tide, which is the principal lunar constituent from the third-degree potential, and the compound constituent $NK_3$ again differ in frequency by 1 cycle in 8.8 y. The compound $NK_3$ is often very small, but it is noticeable at this location because the tidal regime is mixed, with comparably strong tides in both diurnal and semidiurnal bands. Similarly, falling near the compound tide $MO_3$ is a small linear constituent, which is sometimes denoted $F_3$ in honor of Admiral A. Franco who noticed it in records from Cananeia, Brazil; these again require 9 years to separate.

Given the harmonic constants of Table 1, it is straightforward to predict the modulations that will be observed in annual estimates of these four constituents. Consider that any cluster is of the form

$$h = A_0 \cos(\omega_0 t - \varphi_0) + A_1 \cos(\omega_1 t - \varphi_1) + \dots \tag{1}$$

Set $\delta\omega_1 = \omega_1 - \omega_0 \ll \omega_0$, and similarly $\delta\omega_2 = \omega_2 - \omega_0$, etc., so that

$$\cos \omega_1 t = \cos \omega_0 t \cos \delta\omega_1 t - \sin \omega_0 t \sin \delta\omega_1 t$$

with similar expressions for the sine components. If only the first term in $h$ appears, its in-phase and quadrature components are simply $A_0 \cos \varphi_0$ and $A_0 \sin \varphi_0$, respectively. But when the other terms in $h$ are present, we find after gathering terms that the components are:

$$\text{In-phase} = A_0 \cos \varphi_0 + A_1 \cos \varphi_1 \cos \delta\omega_1 t + A_1 \sin \varphi_1 \sin \delta\omega_1 t + \dots$$
$$\text{Quadrature} = A_0 \sin \varphi_0 + A_1 \sin \varphi_1 \cos \delta\omega_1 t - A_1 \cos \varphi_1 \sin \delta\omega_1 t + \dots$$

Evaluating these terms for the four constituents of interest here, using the coefficients tabulated in Table 1, leads to the solid lines in Figure 1. The agreement with the annual estimates (solid circles) is reasonably good, including the extrapolation into the period of the Quarry Bay series (recall Table 1 is based only on North Point data). Thus is explained the major modulations in the tidal constants shown by Devlin et al.

Any physically-based analysis of changes in these tidal constituents should be based on the differences between the annual estimates (solid circles) and their understood modulations (solid lines). The large offset in $N_2$ amplitudes around 2007, and earlier highs around 1994 and 2002, are obvious, and these may represent either true changes in tide or merely instrumental problems. But any analysis of cause must begin with a careful tidal analysis as a first step. Otherwise, computations of tidal correlations are of dubious value.

**References**

Cartwright, D. E. (1975). A subharmonic lunar tide in the seas off western Europe. *Nature*, **257**, 277–280.

Cartwright, D. E. and R. J. Tayler (1971). New computations of the tide-generating potential. *Geophys. J. R. astr. Soc.*, **23**, 45–74.

Devlin, A. T., J. Pan, and H. Lin (2018). Tidal variability in the Hong Kong region, *Ocean Sci. Discuss.*, doi:10.5194/os-2018-62, under review.

Doodson, A. T. (1924), Perturbations of harmonic tidal constants, *Proc. Royal Soc.*, **106**, 513–526.

Munk, W. H. and D. E. Cartwright (1966). Tidal spectroscopy and prediction. *Phil. Trans. R. Soc. London* **259**, 533–581.

---

## Referee Comment (RC1) · Anonymous Referee #1 · 7 Dec 2018

7 December 2018

Comments on 'Tidal variability in the Hong Kong region' by Devlin et al. (OSD)

This paper looks at the variability in the semidiurnal and diurnal tides, and in overtides, around Hong Kong and tries to relate the observed tidal changes to changes over a wider area and in MSL. It is one of a number of papers that have appeared in recent years that have pointed to tantalising associations between changes in tides and MSL that are sometimes enigmatic and always hard to explain.

Therefore, the availability of a large data set from a small region such as Hong Kong is to be welcomed. However, as the authors point out, this region has undergone a lot of engineeering modifications and it is therefore not the easiest of places to try and separate the impacts on the tides from those modifications from those due to genuine changes in large-scale ocean processes (the NW European coastline is a similarly

problematic region given that it has had a lot of dredging etc.). The authors attempt to make that separation by also using data from a small numbers of sites across the vast area of the South China Sea etc. I found that quite unsatisfactory.

The paper seems to me to provide findings which are far from coherent, and so do not lend themselves to easy interpretation. The authors attempt to explain all that diversity by rather (to me) a rambling discussion of 'maybe' processes such as reclamation, changes in baroclinicity, changes in rivers, resonance shift etc. You can explain anything away in this way.

I read the paper several times and my recommendations are:

(i) to rewrite it to focus only on the local data set from the Hong Kong area which, although may be affected by the engineering changes, does seem to present a reasonably spatially coherent set of findings. And then drop the SCS discussion which is superficial at best for such a large area. A local focus, perhaps with some modelling to provide a sensitivity study, would make for a nice paper.

(ii) focus only the four main constituents. The smaller ones can indeed be mentioned in passing (e.g. if M4 is changing in an opposite way to M2) but it is the main ones that most people are concerned with understanding at the moment and, as Ray has pointed out in his interactive comment, it is not clear that the authors properly understand the variability inherent in some of the minor tides and/or in the software used to determine them. I would also drop figures 7-10.

(iii) drop the division of the data set into historical/modern. I found the discussion of the differences between the two epochs unconvincing.

(iv) try and not include so many numbers in the text which the reader just cannnot absorb.

(v) include some mention of changes in tide gauge operations, aside from just whether they were relocated. For example, are some now using radar gauges instead of float

gauges? Have any studies been done of the consequent differences in the tide? Or at leasts flag this as a possible issue.

Some detailed comments:

34 - there is no need for a hyphen in mean sea-level. On the other hand there is in e.g. sea-level rise.

39 - drop 'inter-tidal'

44 - define PSI

48 - well, if you have chaotic results (which are not necessarily the fault of the authors of course), then you can always explain them as a combination of many processes, especially when you have no real data to back up the suggestions. (I know this is a harsh remark, but that's the way this paper reads to me.)

84 - start new sentence at Therefore

96 - +/- 5 percent of what?

97 - 65% ditto

about 97 - the TAC and delta-HAT acronyms are mentioned here but only explained properly below. It seems to assume the reader has read the other Devlin papers. I would define them a little more fully around here.

I don't have a problem with the TAC parameter and name by the way, but I really don't like delta-HAT. As I understand it, it reflects the maximum level that would be obtained in a year from the time-dependent amplitudes and phases extracted from the admittance method? But HAT to most people refers to the maximum level that would be obtained by running a set of tidal predictions over 18.6 years. I would find another name for this parameter. Also it has nothing to do with time series as far as I undertand it, it is just the sum of the amplitudes for either the 4 or 8 constituents for that year (please clarify if not).

98 - doubled. With respect to what? Any exceedance level will be with respect to a datum.

98 - I would drop the TSL acronym. There is no need for too many acronyms. 'Extreme sea level' would do here just as well.

176 - tide gauge records

189 - website should be the website

213 - this is true only if the nodal and other low-frequency modulations (i.e. perigean) are the same in the real ocean as in the potential. There are many examples from shallow-water areas of them not being the same.

223 - state these time series are annual values (presumably)

226 - reword: which has previously been shown to be more apparent

232 - year-to-year change. (See my comment above above delta-HAT which is bad name)

234 - typically 75%

237 - you use the word 'minor' here to refer to N2, K2, P1 and Q1, but minor is used for a different set below. I would change 'minor' here to 'latter four' or similar.

about 244 - I would add 'amplitude' many times in here and in the figure captions. For example, you mention 'tidal perturbations' here - perturbations in what? What are they? I think the problem is the jargon half the time.

251-254 - why is this sentence relevant? You don't do any projections into the future.

265 - say why you use the last 30 years. Data better?

273 - you use the words historical/modern here and early/later lower down which gets confusing. Anyway, as mentioned above, I would drop this aspect.

293 - does 'minor' here mean the 4 above? Be clear.

304 - I am not sure anyone knows where Beibu Gulf is (no offence intended). Perhaps add 'on the south coast of China'.

306/308 - now we have early/later

325 - you quantify the others but not for Bintulu.

392 - a record can be flat or have zero trend. You can't have a 'flat trend'

413 - 'minor' here means quite a different set (discussed by Ray)

417 - there is discussion of the perigean dependence of N2 along the China coast in the Feng et al. paper by the way.

423 - 'missed'. It looks to me to be there is a little bit.

425 - why is this interesting? N2 would be in phase wouldn't it in a small area like this?

456 - 'it is apparent'. It is in figures 7 and 8 ok but not to me for 9 and 10?

464 - who –> which

467 - will be –> are

470 - correlations of what?

659 - the Conclusions for the reasons for the tidal changes are just speculation. You should start this section by reviewing what the data tells you.

824 - it is hard to see the red and green on top of the dark blue. The caption should say the blue shows depth in metres.

figure 3 and others - I read this paper on A4 paper and I cannot read what's in the legends or even the axis annotations of some of the figures.

In (b) and (d) there is a red square box for the Hong Kong area not mentioned in the

caption.

They also have the Egbert model values which are not discussed in the text, so why have them?

In (c) there are captions for each point like CHC which are unnecessary given Figure 1.

figure 5 etc. caption - again the word 'amplitude' needs adding whenever you say something like 'detrended (M2+S2+K1+O1)'.

figure 7 - I can understand the mean values for the tides but the mean values of MSL require to know the datum.

figure 9 and 10 - I can't read the information on the right.

Table 1 - add an extra column for the number of years of data used.

[Figure]

---

## Referee Comment (RC2) · Anonymous Referee #2 · 3 Jan 2019

The authors set out to investigate how the observed tides around Hong Kong, and in the wider SCS, have changed over the last decades. The use of such a large data set from a small region is interesting, and there are some intriguing results, but there are issues I think must be addressed before this could be published. Both of these points are already raised by Review 1 and by Richard Ray in their comments, and I second them here (hence the brevity of this review).

Major comments The paper is a difficult read, mainly because we are constantly interrupted by quantifications. The reader could look up numbers in the figures and tables rather than being told that this gauge changed this much compared to that gauge. Maybe consider saying that "A increased more than B with a factor N". The overtide analysis really doesn't add much, even if it wasn't flawed (see Ray's comment). If it is to be included, and I don't think it will be significant once it is analysed properly, we will have to be told why the changes are of interest. I think it would be more worthwhile,

and this is seconding Review 1, to focus on the main constituents around Hong Kong alone, and delete the speculations about why the tides may have changed in the SCS. If the latter part is to be included, we need to be told with more certainty why these changes have occurred.

Minor comments L127: it is surprising to not see references to work by Alford and collaborators here.# L176-196: I suggest deleting this and just give a very brief summary: we have NN gauges spanning NN years (see table and figures…). L273: why distinguish between historical and modern, using some arbitrary cutoff? Technically, they are all historical, since they are in the past…

---

## Author Comment (AC1) · 27 Feb 2019

7 December 2018

Comments on 'Tidal variability in the Hong Kong region' by Devlin et al. (OSD)
This paper looks at the variability in the semidiurnal and diurnal tides, and in overtides,
around Hong Kong and tries to relate the observed tidal changes to changes over a
wider area and in MSL. It is one of a number of papers that have appeared in recent
years that have pointed to tantalising associations between changes in tides and MSL
that are sometimes enigmatic and always hard to explain. Therefore, the availability of a large
data set from a small region such as Hong Kong is to be welcomed. However, as the authors
point out, this region has undergone a lot of engineering modifications and it is therefore not
the easiest of places to try and
separate the impacts on the tides from those modifications from those due to genuine
changes in large-scale ocean processes (the NW European coastline is a similarly problematic
region given that it has had a lot of dredging etc.). The authors attempt to make that
separation by also using data from a small number of sites across the vast area of the South
China Sea etc. I found that quite unsatisfactory. The paper seems to me to provide findings
which are far from coherent, and so do not lend themselves to easy interpretation. The
authors attempt to explain all that diversity by rather (to me) a rambling discussion of 'maybe'
processes such as reclamation, changes in baroclinicity, changes in rivers, resonance shift etc.
You can explain anything away in this way.

*-Thank you for your review, and for your constructive comments! We are thankful that you
recognize that our study is of interest. We are also thankful to have a critical eye to evaluate
our results. As we think is clear to you, we attempted to do far too much in this study! We
have been studying these tides gauges for quite a while now and have found quite a bit of
interesting behaviour that has piqued our curiosity in many ways. In the course of writing
this first draft, we tried to include everything that we had observed, even those things that we
were not yet quite sure of (such as the "minor tides" tangent that has now been better
elucidated to us by the comment of Richard Ray as an error in analysis approach), or things
that are hard to make conclusions about (such as trying to determine anything meaningful
about the SCS tides relying solely on the sparse and only historical publicly available
Mainland China tide gauge network). We admit that we tried to be far too ambitious in this
first attempt. Furthermore, this paper has been waiting for review for a very long time, and
while we have been waiting, we have moved forward in our work and found new discoveries
and methods that have provided new insights about the data in HK. For example, Richard
Ray's comments led us to the correct way to analyse minor tides such as $M_3$ and $N_2$ (i.e., use
a 9-year window for analyses) which produced stable results without the 9-year "pseudo-
cycle" from constituent contamination. However, this focus on minor constituents, while
improved, is not very relevant to the overview of tidal correlations in Hong Kong, which is
focused on yearly-scale fluctuations that are not as apparent after performing 9-year
analyses. This part of the analysis is saved for a future, global-based study of $M_3$ based on 9-
year analyses.*

*Based on your comments, Richard Ray's comment, and the new things we have been
working on, we have now greatly streamlined this paper and made a better focus on relevant*

*and communicable results. The major changes are that we have dropped many things in the first draft and made the focus the Hong Kong local results of the four largest tides.*

*The relevant omissions are:*
*-The "minor tide" analyses (i.e., $N_2$, $K_2$, $Q_1$, and $P_1$) and consequently the delta-HAT-8 analyses.*
*-The South China Sea results and discussion. Also, much of the related introduction materials about the SCS dynamics, internal tide generation and propagation, etc.*
*-The "historical" vs. "modern "comparisons.*
*-The later discussion about $M_3$ and other minor tidal behaviour (this part was erroneous as pointed out by Ray).*
*-Figures related to the above, which has allowed a better resolution to be used without "tiling" the results and making them too small.*
*-Removal or downplaying of the suggestion of mechanisms to explain the behaviour, besides some short mentions of the possible importance of engineering projects in HK. This possibility will be explored in an upcoming modelling study using highly accurate DEMs*

*-Targeted responses to your individual comments are found below. Many of these comments are not applicable after we removed the majority of things listed above.*

*-We will do our best to reference our relevant changes in relation to the original text line numbering and sectioning. However, with all the omissions, the form and structure of the text has greatly changed and referring to the old numbering will likely be confusing. Therefore, we will describe the changes in reference to new line numbering where applicable.*

"this region has undergone a lot of engineering modifications and it is therefore not the easiest of places to try and separate the impacts on the tides from those modifications from those due to genuine changes in large-scale ocean processes."

*-In regard to this comment, in the new version, we have excluded a lot of the hypothesizing about what is causing the tidal changes (i.e., local vs. regional mechanisms) and instead just mention that coastal modifications have had a long history in HK and may be at least partly to blame via possible resonance and frictional changes. And, as we are no longer including or talking about any of the SCS observations, we don't think it is needed to make any substantial hypothesizing about the regional tidal properties.*

I read the paper several times and my recommendations are:

(i) to rewrite it to focus only on the local data set from the Hong Kong area which, although may be affected by the engineering changes, does seem to present a reasonably spatially coherent set of findings. And then drop the SCS discussion which is superficial at best for such a large area. A local focus, perhaps with some modelling to provide a sensitivity study, would make for a nice paper.

*-Thank you for the feedback and suggestion. We have followed this advice and have now focused only on the Hong Kong results. The SCS discussion has now been omitted, as this*

*data is sparse and historical. Since this dataset is mainly composed of Mainland China observations, which have not been publicly updated since 1997, studying this data does not really reveal anything useful, even though I really hoped that I could have. We had the best of intention in using this data, hoping that a discussion of these results might help fuel an interest in releasing more data publicly, but we admit this case has not been made.*

*-As to the suggestion about modelling, we decided to not undertake such an endeavor here. It is believed that capturing the full dynamics of the HK waters is a complex question and will take a lot of careful consideration of details (such as highly-accurate DEM that can simulate the differences in tides under different land reclamation projects of the past) and will hopefully be the subject of future studies. This present study is only meant to be an observational study to identify the interesting tidal observations. While we believe that is highly likely that the coastal modifications have something to do with this, it is also believed that proving this via modelling would be worthy of a completely different study, which we do hope to pursue more completely soon.*

(ii) focus only the four main constituents. The smaller ones can indeed be mentioned in passing (e.g. if M4 is changing in an opposite way to M2) but it is the main ones that most people are concerned with understanding at the moment and, as Ray has pointed out in his interactive comment, it is not clear that the authors properly understand the variability inherent in some of the minor tides and/or in the software used to determine them. I would also drop figures 7-10.

*-Thank you again for the comments which have focused our scope. We do indeed now only discuss the four major constituents ($M_2$, $S_2$, $K_1$, and $O_1$), and the delta-HAT based on these four tides. The other four major tides ($N_2$, $K_2$, $P_1$, and $Q_1$) do not add much to the discussion here, and the delta-HATs based on 8 tides was not too much different from the four-tide rendering, so it is better to focus only on the four most stable tides. We have also removed the old versions of Figure 8-10, which were too noisy and mostly useless, and, as illustrated by Ray, are erroneous in approach. However, we have elected to keep Figure 7(now Figure 9) which shows the major tidal anomalies witnessed at Quarry Bay and Tai Po, and now have a briefer discussion about these observations in the context of timing with major reclamation projects as a motivation for future modelling efforts.*

(iii) drop the division of the data set into historical/modern. I found the discussion of the differences between the two epochs unconvincing.

*-Agreed. We were attempting to make something meaningful out of the sparse Mainland China data in relation to the HK data, but this attempt was unsuccessful. Some brief mentions are still included about the fact that there is an obvious difference in tidal variability in early years and later years at the longer records, which was another reason we decided to keep the (old) Figure 7 (now Figure 9).*

(iv) try and not include so many numbers in the text which the reader just cannot absorb.

*-Thank you for this comment. We agree that too many numerical results in the text can make a boring "laundry list" of data that is too hard to read. We hope that this issue has now been alleviated by the removal of the SCS data and the historical/modern comparisons.*

(v) include some mention of changes in tide gauge operations, aside from just whether they were relocated. For example, are some now using radar gauges instead of float gauges? Have any studies been done of the consequent differences in the tide? Or at least flag this as a possible issue.

*-Thank you for this comment. The QB gauge is the only one that was ever re-located, and to the best of our knowledge, there are no known discrepancies or errors have been documented at any gauges. All other gauges have had settlement measurements made since 1991, with no significant changes observed. We have been closely working with the data provider, the Hong Kong Observatory, at the senior level, who are well-versed in the history and quality control of the data and can verify the quality of all of data. All gauges are currently radar gauges. We therefore believe that there are no major datum issues, instrumentation issues, or other errors in our set of data used here. We have also added a few ore publications that are old official reports from HKO about early tidal analyses of the HK tide gauge network.*

Some detailed comments:

34 - there is no need for a hyphen in mean sea-level. On the other hand there is in e.g. sea-level rise.

*-Thanks for the clarification. We have fixed these instances here and elsewhere.*

39 - drop 'inter-tidal'

*-Dropped, thanks.*

44 - define PSI

*-Done*

48 - well, if you have chaotic results (which are not necessarily the fault of the authors of course), then you can always explain them as a combination of many processes, especially when you have no real data to back up the suggestions. (I know this is a harsh remark, but that's the way this paper reads to me.)

*-Thanks for the comment, and no offense taken. We admit this was a bit rambling. We now have focused this better to be applicable to HK, and only suggest the possibility of the frictional/resonance mechanism under rising MSL because of local engineering changes.*

84 - start new sentence at Therefore

*-Done.*

96 - +/- 5 percent of what?

*-This indicates a 5% modification of total sea level due to tides for an arbitrary MSL change. We have tried to make the language clearer.*

97 - 65% ditto

*-This number is dropped, and the discussion is better focused now.*

about 97 - the TAC and delta-HAT acronyms are mentioned here but only explained properly below. It seems to assume the reader has read the other Devlin papers. I would define them a little more fully around here.

*-I have tried to explain these metrics better here, or at least enough to be introduced here, with the details better discussed in Methods..*

I don't have a problem with the TAC parameter and name by the way, but I really don't like delta-HAT. As I understand it, it reflects the maximum level that would be obtained in a year from the time-dependent amplitudes and phases extracted from the admittance method? But HAT to most people refers to the maximum level that would be obtained by running a set of tidal predictions over 18.6 years. I would find another name for this parameter. Also it has nothing to do with time series as far as I understand it, it is just the sum of the amplitudes for either the 4 or 8 constituents for that year (please clarify if not).

*-Thank you so much for this comment. We will try to answer this carefully and explain our logic. Over the course of developing these novel methods in other studies (Devlin et al, 2014; 20171; 2017b), we wrestled with many different acronyms and names for out metrics of what is now TACs and delta-HATs. For instance, originally, we called them TAT (tidal anomaly trends) in Devlin et al., 2014, but later decided that name was inaccurate, as what we observe is not really a "trend". So, we decided that TAC was better later (Devlin et al., 2017). But I felt a little conflicted about changing the acronym, since it had already been established in my previous study. A similar situation applied to the use of delta-HAT. At the onset, my co-authors and I acknowledged that some people would think of the classical definition of HAT (based on the 18.6-year analysis). We decided that using "delta-HAT" would imply a shorter timescale change in this metric, which could not be revealed from an 18.6 yr analysis; we are interested in yearly-scale fluctuations. However, we have always introduced our method as a "proxy" or "indirect estimate" of the change in HAT. Since this language and acronym has been used in a recent paper (Devlin et al., 2017a), as well as in a new paper that studies the Atlantic using these methods which was recently accepted, we really want to keep the language consistent in the current paper about Hong Kong.*

*To clarify our methods, we do combine the tidal amplitudes and phases of the top four tides garnered from the yearly admittance values into a single complex time series, and the absolute value is taken to show the highest actual level reached by that combination, which is then detrended and regressed against detrended MSL over the same window.*

*We therefore want to make the case that we want to keep this name of delta-HAT, but we will also better explain the distinction of it being a "proxy" in the manuscript where applicable. If you still take issue with the use of this acronym, we will relent and try to find a new one, especially if you have a good suggestion.*

98 - doubled. With respect to what? Any exceedance level will be with respect to a datum.

*-Doubled, as in almost double of the exceedance of MSL alone (above an arbitrary datum).*

98 - I would drop the TSL acronym. There is no need for too many acronyms. 'Extreme sea level' would do here just as well.

*-We have adopted this format now.*

176 - tide gauge records

*-Fixed.*

189 - website should be the website

*-Fixed.*

213 - this is true only if the nodal and other low-frequency modulations (i.e. perigean) are the same in the real ocean as in the potential. There are many examples from shallow-water areas of them not being the same.

*-Good point, and we have added this caveat to the methods. However, every gauge used has been carefully analysed by eye to identify any instances of "leakage" of low-frequency signals (nodal or otherwise). As Ray has instructed in his comment, the 8.85 yr perigean cycle can still be apparent after admittance methods are applied in $N_2$ and other constituents. But we have now dropped the $N_2$ analysis from the paper. However, in the HK region, the low-frequency modulations of the four main tides are not apparent after the admittance method is applied.*

223 - state these time series are annual values (presumably)

*-We have been more explicit in this paragraph about time-series being annual.*

226 - reword: which has previously been shown to be more apparent

*-Fixed, thanks!*

232 - year-to-year change. (See my comment above about delta-HAT which is bad name)

*-We have made this change. However, please see the comment above about the use of the delta-HAT name.*

234 - typically 75%

*-Fixed*

237 - you use the word 'minor' here to refer to N2, K2, P1 and Q1, but minor is used for a different set below. I would change 'minor' here to 'latter four' or similar.
about 244 - I would add 'amplitude' many times in here and in the figure captions. For example, you mention 'tidal perturbations' here - perturbations in what? What are

they? I think the problem is the jargon half the time.

*-Most of these comments are no longer relevant since I removed a lot of material about minor tides, but I will clarify that we use "tidal perturbations" to indicate the variation of the tidal admittance from the (detrended) mean value.*

251-254 - why is this sentence relevant? You don't do any projections into the future.

*-This sentence did not mean to assume anything about projections. We include this part to clearly indicate that the calculated TACs can be assumed stable and constant over the time window considered (30 years or less), a "pseudo-linear" assumption, if you will. This point needed to be stressed in previous papers, and this window length was employed in all previous works by Devlin et al. so far. Mainly it is matter of consistency (similar to our justification of keeping the delta-HAT moniker). Comparative analyses in other studies have shown that any longer of a regression analysis may obscure changes in the TAC over time. However, we have modified the text a bit to reflect this explanation, and removed the word "extrapolate" which may have been adding to the confusion.*

265 - say why you use the last 30 years. Data better?

*-Please see the above comment. To reiterate, longer time window may obscure changes in the TAC values, and 30 years has been shown to be a good window to use based on previous studies.*

273 - you use the words historical/modern here and early/later lower down which gets confusing. Anyway, as mentioned above, I would drop this aspect.

*-These comments should no longer be applicable, as the historical and minor material has been omitted now.*

293 - does 'minor' here mean the 4 above? Be clear.

*- These comments should no longer be applicable*

304 - I am not sure anyone knows where Beibu Gulf is (no offence intended). Perhaps add 'on the south coast of China'.

*-Some in America and the Western world would know it as the Gulf of Tonkin. I was trying to use the name that the locals use. However, this comment is now moot since I have removed the SCS discussions.*

306/308 - now we have early/later

*-No longer included.*

325 - you quantify the others but not for Bintulu.

*-No longer relevant.*

392 - a record can be flat or have zero trend. You can't have a 'flat trend'

*-Fixed.*

413 - 'minor' here means quite a different set (discussed by Ray)

*-No longer relevant.*

417 - there is discussion of the perigean dependence of N2 along the China coast in the Feng et al. paper by the way.

*-Thanks for the reference. This is no longer relevant to the current paper, but I am doing a new study about the perigean-modified tides based on a world-wide set of data using 9-year analyses, so I will read that paper with great interest!*

423 - 'missed'. It looks to me to be there is a little bit.

*-No longer relevant; material removed.*

425 - why is this interesting? N2 would be in phase wouldn't it in a small area like this?

*-No longer relevant; material removed.*

456 - 'it is apparent'. It is in figures 7 and 8 ok but not to me for 9 and 10?

*-No longer relevant; material removed.*

464 - who –> which

*-No longer relevant; material removed.*

467 - will be –> are

*-No longer relevant; material removed.*

470 - correlations of what?

*-No longer relevant; material removed.*

659 - the Conclusions for the reasons for the tidal changes are just speculation. You should start this section by reviewing what the data tells you.

*-Thanks for this comment. We have done a lot of work to rewrite the conclusion section with some more focus. We now try to simply discuss the observations, discus lightly some possible reasons (i.e., harbour changes or regional climate changes), and lay out some future possible steps.*

824 - it is hard to see the red and green on top of the dark blue. The caption should say the blue shows depth in metres.

*-No longer relevant; maps of the SCS are now removed.*

figure 3 and others - I read this paper on A4 paper and I cannot read what's in the legends or even the axis annotations of some of the figures.

*-Figure have all ben redone in the new version.  Without the SCS material, we now use single-panel figures, focusing only on the most relevant observations.*

In (b) and (d) there is a red square box for the Hong Kong area not mentioned in the caption. They also have the Egbert model values which are not discussed in the text, so why have them?

*-No longer relevant; material removed.*

In (c) there are captions for each point like CHC which are unnecessary given Figure 1.

*-OK, we now only show station names in Figure 1, and all other figures have names removed.*

figure 5 etc. caption - again the word 'amplitude' needs adding whenever you say something like 'detrended (M2+S2+K1+O1)'.

*-Thanks, fixed!*

figure 7 - I can understand the mean values for the tides but the mean values of MSL require to know the datum.

*-All water levels given for MSL are in relation to the "chart datum" as defined by the Hong Kong Observatory. The chart datum is defined as an additional 0.146 m below the Hong Kong Principal Datum (HKPD).  The HKPD determined for the years 1965-1983 was approximately 1.23 m below MSL.  The HKPD has been recently re-determined using data from 1997-2015 to be 1.30 m below MSL.  Therefore, all MSL values are given in relation to the sum of both values, so 1.376 m for the early years, and 1.446 m for the later years (approximately).  Please see the following weblink for a full history of the datum in HK: [https://www.hko.gov.hk/blog/en/archives/00000204.htm](https://www.hko.gov.hk/blog/en/archives/00000204.htm).  However it should be mentioned that what is shown in current figure, what is plotted for the MSL component is the zero-frequency component of the harmonic analysis (i.e., de-tide MSL) which may have some offset from the full MSL (tides included) as determined from HKO.  We now include more material about this history in the text.*

figure 9 and 10 - I can't read the information on the right.

*-No longer relevant; material removed.*

Table 1 - add an extra column for the number of years of data used.

*-OK, done!*

---

## Author Comment (AC2) · 27 Feb 2019

The authors set out to investigate how the observed tides around Hong Kong, and in the wider SCS, have changed over the last decades. The use of such a large data set from a small region is interesting, and there are some intriguing results, but there are issues I think must be addressed before this could be published. Both of these points are already raised by Review 1 and by Richard Ray in their comments, and I second them here (hence the brevity of this review).

*-Thank you for your overview and gentle review. We have worked hard in this revision to make some major changes, based on the very helpful comments of Reviewer #1 and the additional helpful comments and explanations added by Richard Ray. It has also been a long time since our initial submission of this paper (we were waiting for an initial review for over 6 months), and a lot of new work and new discoveries are evolving in our examination of the HK waters.*

> *In this version, we have removed a lot of the contentious material. We now only use the four major tidal constituents and the sum of the four as the delta-HAT. We also remove the overtide analyses, the SCS analyses and discussion, and the historical/modern comparisons. Finally, we have removed the "minor tide" discussion of the $M_3$ tide and other lesser components, because, as pointed out by Ray, this effort was somewhat flawed in execution. Richard's comments were quite helpful in elucidating the proper method of analysing perigee-influenced tides (such as $M_3$ and $N_2$), in that a 9-year analysis window should be employed. In fact, we are moving forward in a new study to examine the global occurrence of the $M_3$ tide using 9-year analysis windows, and these results are quite interesting, including the HK results. However, these results are now not as relevant to the current study, so all previous material about this has been removed. We now focus only on the local HK results, and downplay talking too much about the mechanisms why, though it is still hypothesized that the local harbor changes are likely part of the answer. However, regional SCS changes under climate change may also be a factor, and, as Reviewer #1 points out, it is difficult to separate the local engineering changes from regional climatic changes. Therefore, we try not to suppose or speculate too much here, mainly we just report what is observed. We do intend to design a new modelling study in the near future that will employ a highly-accurate DEM of HK and apply some of the historical coastal changes to examine if tidal properties can be affected by such changes, and we mention this intention for the future in the new manuscript. Please also see our complete responses to Reviewer #1 for additional responses and explanations.*

*The relevant omissions are:*
> *-The "minor tide" analyses (i.e., $N_2$, $K_2$, $Q_1$, and $P_1$) and consequently the delta-HAT-8 analyses.*
> *-The South China Sea results and discussion. Also, much of the related introduction materials about the SCS dynamics, internal tide generation and propagation, etc.*
> *-The "historical" vs. "modern "comparisons.*
> *-The later discussion about M3 and very minor tidal behaviour (this part was erroneous as pointed out by Ray).*
> *-Figures related to the above, which has allowed a better resolution to be used without "tiling" the results and making them too small.*

*-Removal or downplaying of the suggestion of mechanisms to explain the behaviour, besides some short mentions of the possible importance of engineering projects in HK. This possibility will be explored in an upcoming modelling study using highly accurate DEMs*

Major comments:

The paper is a difficult read, mainly because we are constantly interrupted by quantifications. The reader could look up numbers in the figures and tables rather than being told that this gauge changed this much compared to that gauge. Maybe consider saying that "A increased more than B with a factor N".

*-Thank you for the comments. We agree that the previous version was a bit confusing and had too many numbers to talk about. We hope that most of these concerns will have been alleviated by the removal of the SCS material, the minor tides, and the historical/modern comps. Beyond this, we have rewritten the remainder of the paper with a careful eye on giving a smooth dialogue without too many quantification interruptions.*

The overtide analysis really doesn't add much, even if it wasn't flawed (see Ray's comment). If it is to be included, and I don't think it will be significant once it is analysed properly, we will have to be told why the changes are of interest. I think it would be more worthwhile, and this is seconding Review 1, to focus on the main constituents around Hong Kong alone, and delete the speculations about why the tides may have changed in the SCS. If the latter part is to be included, we need to be told with more certainty why these changes have occurred.

*Thank you once again. We agree and have now removed the overtide discussion. We now focus only on the largest and most familiar four tides and have downplayed most of the discussion that speculates about the reasons why. As mentioned above, we do hypothesize that the harbour changes may be at play, but without any modelling or better explanations (which may be available after future studies are completed), it is difficult to explain the relative importance of local and regional changes to changing tides.*

Minor comments:

L127: it is surprising to not see references to work by Alford and collaborators here.

*-Thanks for the suggestion. We did in fact read some of Alford's papers and did have some references included in earlier versions of the manuscript. However, we edited many times and removed a lot of this discussion that was determined to not be as relevant to the new direction of the manuscript and had somehow removed a few papers that were done by Alford and other collaborators in the text, though they are still listed in the Reference list (e.g., Alford, 2008; Chinn et al., 2012). We have re-evaluated these sections, and now cite these two papers in the appropriate place now.*

L176-196: I suggest deleting this and just give a very brief summary: we have NN gauges spanning NN years (see table and figures: : :).

*-Thanks, we have now honed down this section to be brief according to your suggestion.*

L273: why distinguish between historical and modern, using some arbitrary cutoff? Technically, they are all historical, since they are in the past.

*-Thanks for the comment. We agree and have removed the comparisons of historical/modern times.*

---

## Author Comment (AC3) · 27 Feb 2019

(from Richard D. Ray, *NASA Goddard Space Flight Center, Greenbelt, Maryland, USA*)

Dear Dr. Ray,

*We would like to thank you greatly for your help and insight provided by your comment. You recognized that we had made some errors in our approach and our logic in analysing the minor tidal constituents in the Hong Kong waters, yet you provided a clear and kind elucidation of what was done incorrectly as well as teaching us the fine-scale details of these factors that have increased our knowledge and understanding. Your discussion has also steered us to perform better analyses, as well as opening up some new inspiration.*

*The result of your discussion coupled with the suggestions of the anonymous reviewers convinced us that the analyses and discussion of "minor" tides should be removed from the current paper. Now, this paper only focuses on the four largest tides; the other leaser tides and overtides are not included now, and neither are the contrasts between Hong Kong and the SCS tide gauges nor the "historical" vs. "modern" comparisons. If you can see my responses to the other reviewers, you should be able to see a better discussion of all the changes and the reason why. In general, our paper is now much shorter and much more focused on defensible and clear results in the Hong Kong waters.*

*However, even though we have removed all the material that your original comment addressed (and therefore we think that the current manuscript will have no conflict with the original comments), we want you to know that your insight has guided us to working on a new study that is looking specifically at the $M_3$ variation, analysed throughout the world ocean. In this new study, we follow your steps and perform 9-year analyses on the tide gauge records. This results in values for $M_3$ and other minor tides that removes the ~8.85 year variability seen before, and yields a more constant*

*and defensible result. We observe from comparing the M$_3$ tide done with 1-year and ~9-year analyses to see that the mean value is similar, but the ~9-year oscillation is no longer present. Interesting, though, some variability at other time scales is still present. For your consideration we show a simple plot of the results in Hong Kong (Quarry Bay), with the 9-year HA shown as a blue line (overlapping 9-year analyses, at one month steps), and the previous 1-year analyses shown as a broken grey line.*

[Figure]

*This is a rough plot (please disregard the spurious data at the end of the record, but the basic form is clear. We have also been performing similar 9-year analyses at every gauge in the world ocean where a significant M3 tide has been observed and hope to soon make a paper that discusses the data and the possible reasons, likely aided by altimetry data.*

*This work is ongoing, but it would not have been possible without your help. We wish to thank you once again for your time and effort with your comments. When we finish our new study, I hope that you will approve! In regards to the current study about Hong Kong, we believe that since no*

*minor tides are now discussed, nothing contentious should remain in this manuscript.*

---

## Referee Report (RR1)

April 2019

Comments on resubmission of 'Tidal variability in the Hong Kong region'
by Devlin et al. (Ocean Science)

This paper is a resubmission of an earlier and longer paper submitted to OSD
which discussed tidal changes in the Hong Kong region, and less convincingly, in
the South China Sea. I am pleased that the authors considered my suggestion to
shorten it and focus on Hong Kong.

I have read it again carefully and I have no doubt that the analysis has been
done well at a technical level. However, my main concern is that the text does
not read well at all. I have made some suggestions on rewording below.

The second concern, which I mentioned last time, is that the parameters used
(TACs and delta-HATs) are simple ones but they are non-standard in tidal
literature (unless you are familiar with Devlin's previous papers). They have to
be explained therefore. But the paper assumes that the reader has read, or now
wants to read, the previous papers. I insist that the paper have an Appendix
wherein these two parameters are explained adequately.

As I mentioned last time, I don't have a problem with the TAC parameter.
However, I really don't like the name 'approximate delta-HAT' which goes against
all common use of the term 'HAT' in tidal literature. As I understand it, it
reflects the maximum level that would be obtained in a year from a chosen set of
time-dependent amplitudes and phases extracted by the admittance method. So
please spell this out in the Appendix.

Detailed comments, many trivial to do with the text:

.. tidal variability in the 31-year period 1986-2016.

40/41 ... locations, time series of approximations of the parameter delta-HAT,
computed from combinations of the major tidal constituents, are found to be
highly sensitive ..

- individual tides --> individual tidal constituents

- as important --> important in combination with

I really don't think tidal changes will ever be as important as MSL rise but as
you say the tidal changes will add to the problem

- additional shorter-term

- and would imply that flood risk

- considered as a substantial complement to

- critical interest if all such factors are undergoing change.

At this point you should refer to the Appendix.

- as a proxy for what can be described as changes ..

- was seen mmm should be mm m

- ok so you have tidal changes and MSL in the extremes, what about non-tidal
and non-MSL changes like storm surges?

- I have never seen the word metropolises used in English (although it is correct I think). I would replace 'urban metropolises' with 'areas'

two extensives in this sentence

SCS --> South China Sea (SCS)

You define this acronym only at the moment in figure 2 caption

114-116 - this sentence needs rewording. It reads like you do something by doing the something.

- the longest record in Table 1 is 1954-2016 which is 63 and not 65 years

- you have not yet explained to the reader that station names are accompanied by station codes, so I would drop (QB) here and see below.

environment --> geographical setting

.. including station name and station code, latitude .... and the ranges of the data records used in this study.

Then add here:

For brevity, we often refer to stations below by their station codes rather than their full names e.g. QB for Quarry Bay.

You use station codes a lot in the text which seems unnecessary to me when it would be much clearer to the reader to use the place names. There is no space shortage here. So, if you were to remove them all and replace with the names then the above sentence would not be needed.

133-137 - I know what you mean here but this sentence is rather a mouthful. Can you split into two? Also the sentence at line 143 'The tidal potential' should come earlier.

- 'effects are eliminated'. Why? I don't understand this. They would be eliminated only in an analysis of 18.6 years and would well and truly still be present in a 1-year analysis.

- 'may not always hold true'. You could refer to Amin's papers.

start a new para at 'The result'

-.. analysis window (e.g. at mid-year) ..

- more apparent in the data sets used here

- ... MSL variability (Appendix 1). With the use of the TACs we determine ...

.. (delta-HAT) (see Appendix 1).

Start a new para.

- mmm should be mm m

- The use of a window of a year in a harmonic analysis

181-183 This sentence could come earlier where you mention the Appendix start new para at 'For the'

Here you start using codes in the text. I have no idea what TPK means and I can't be bothered referring all the time to Figure 1

years should be 31 (1986-2016)

.. determinations (Table 1).

.. in the TAC values over time .. [although I am not sure I understand this. I would reword this and simply say that as the TAC could change over time you have adopted a common epoch for the work]

twiddle 12-30 --> 12-31 (Table 1).

- I was not provided with the supplement

- I believe Victoria Harbour is usually spelt with a 'u'. It would be good to show it on Figure 1.

.. all other gauges except .. moderately negative ...

- you add the acronyms in the header at line 200 so do the same in the header here discrete --> particular drop 'In Hong Kong' Five stations ...

- HK --> Hong Kong.

- drop 'and we report ..'. Irrelevant.

- OT --> overtides (OT)

drop 'an additional'

241-242 reword: .. Therefore, all MSL values reported here are given relative to the HKPD for the epoch 1965-1985.

drop 'drastically'

correlated to --> correlated with though --> although

- I don't understand the sentence 'The TACs'. What does it mean that they are present? You mean they are large or what? And you don't actually show delta-HATs but you do show the TACs of the approximate delta-HATs. This all needs rewording.

The spatial similarity in the ..

- with processes at other frequencies, such as at varies with with the spring-neap drop 'away'

forcing of the tides

- some records are of shorter length and/or have many gaps, making ..

as was perform --> employ three-dimensional numerical ocean models to simulate the changing impacts on comma before 'to better'

As I mentioned last time, one limitation of this study is the possibility of instrumental changes in the tide gauges. You don't even mention what sort of gauges they are or what changes there might have been.

drop hyphen in sea level. There should be a hypen only when used as an adjective e.g. sea-level rise.

can be positively reinforced by what? by MSL changes?

agitate --> aggrevate

360-364 these web addresses should have http or https interest --> interests

383/422 - you could just call it SCS if has been defined in the text figure 1/2 - add extra names as mentioned above figure 2 - red on dark blue is not good. I would add China.

Taiwan Strait and Luzon are very small and unreadable when printed in A4

Figure 3 etc. - mention again that red/blue is +/-.

Black marks indicate TACs which are not significantly different from zero.

figure 7 - as I understand it this is not delta-HAT but the TAC computed from the approximate delta-HAt time series made from 4 constituents. Right? Please reword the caption to make that clear and not read as jargon. Also you have not defined what delta-HAT4 means in the text.

sea-level --> sea level giving the station names and station codes, ... year of the available records, as well as the range of data analysed ..

- and O1 over the period 1986-2016.

- TACs over the period 1986-2016.

TAC --> TACs

---

## Author Response (AR2)

Topic Editor Decision: Publish subject to minor revisions (review by editor) (10 May 2019) by John M. Huthnance

Comments to the Author:

Dear Authors

Thank-you again for your revised manuscript. Both referees have seen it and as a result I am asking for minor modifications prior to publication in Ocean Science.

I am not sure whether you have seen the comments so I am copying them below with a few "editorial" comments from myself. Most are from Referee 1 who is an expert in tides (as well as a "native" English speaker) so please do address these. Referee 2 endorses a couple of points. There is also a comment about figure 9 forwarded via Referee 1 (you might guess where it comes from).

Referee 1

Comments on resubmission of 'Tidal variability in the Hong Kong region'

by Devlin et al. (Ocean Science)

This paper is a resubmission of an earlier and longer paper submitted to OSD which discussed tidal changes in the Hong Kong region, and less convincingly, in the South China Sea. I am pleased that the authors considered my suggestion to shorten it and focus on Hong Kong.

*-Thank you very much for all your constructive comments in the previous revision!*

I have read it again carefully and I have no doubt that the analysis has been done well at a technical level. However, my main concern is that the text does not read well at all. I have made some suggestions on rewording below.

*-Thank you for these new comments, we will pay close attention to these suggestions as well as giving all parts of the manuscript a careful edit for readability.*

The second concern, which I mentioned last time, is that the parameters used (TACs and delta-HATs) are simple ones but they are non-standard in tidal literature (unless you are familiar with Devlin's previous papers). They have to be explained therefore. But the paper assumes that the reader has read, or now wants to read, the previous papers. I insist that the paper have an Appendix wherein these two parameters are explained adequately.

*-OK, we have added an appendix to better explain these terms. Thank you for your patience and understanding. As mentioned previously, we realize now that the term "delta-HAT" may not have been the best choice as it may confuse some people, but we still wanted to try and keep consistent with previous papers. We hope that our logic in the Appendix will be able to satisfy all readers.*

As I mentioned last time, I don't have a problem with the TAC parameter. However, I really don't like the name 'approximate delta-HAT' which goes against all common use of the term 'HAT' in tidal literature. As I understand it, it reflects the maximum level that would be obtained in a year from a chosen set of time-dependent amplitudes and phases extracted by the admittance method. So please spell this out in the Appendix.

*-Yes, absolutely! We have carefully thought about the definition and meaning of our δ-HAT terminology and have also endeavored to explain previous definitions of HAT as compared to our "δ-HAT". We hope our efforts are now easier to understand!*

Detailed comments, many trivial to do with the text:

.. tidal variability in the 31-year period 1986-2016.

*-Done.*

40/41 ... locations, time series of approximations of the parameter delta-HAT, computed from combinations of the major tidal constituents, are found to be highly sensitive ..

*-Fixed*

- individual tides --> individual tidal constituents

*-Fixed*

- as important --> important in combination with

*-Fixed*

I really don't think tidal changes will ever be as important as MSL rise but as you say the tidal changes will add to the problem

*-Agreed, thanks for the input.*

- additional shorter-term

*-Fixed*

- and would imply that flood risk

*-Done*

- considered as a substantial complement to

*-Fixed*

- critical interest if all such factors are undergoing change.

*-Done*

At this point you should refer to the Appendix.

*-Ok, we refer to the Appendix here.*

- as a proxy for what can be described as changes.

*-OK, fixed.*

- was seen

*-OK, fixed.*

mmm should be mm m

*-OK.*

- ok so you have tidal changes and MSL in the extremes, what about non-tidal and non-MSL changes like storm surges?

*-Thanks, this is a good comment. We have added a bit of text here to explain better that storm surge was not considered in the cited study here (or in the present study), though we do say that since tides and storm surge are both long-wave processes, they may be due to similar reasons, and knowledge of one part of the water level spectrum may instruct about other parts of the spectrum.*

- I have never seen the word metropolises used in English (although it is correct I think). I would replace 'urban metropolises' with 'areas'

*-Ok, I have changed this. Actually, I have also seen the word "Megalopolis" used to describe the ultra-populated areas of the planet (such as the Pearl River Delta which includes approx. 100 million people. But we do not use this term here.*

two extensives in this sentence

*-We changed the 2$^{nd}$ instance.*

SCS --> South China Sea (SCS)

*-OK, fixed!*

You define this acronym only at the moment in figure 2 caption

*-Should be fixed by the previous comment.*

114-116 - this sentence needs rewording. It reads like you do something by doing the something.

*-I have now cut down this sentence to be more direct and clearer.*

- the longest record in Table 1 is 1954-2016 which is 63 and not 65 years

*-Fixed.*

- you have not yet explained to the reader that station names are accompanied by station codes, so I would drop (QB) here and see below.

*-OK, I agree with your comments below about explaining station codes, so we will go forward with this explanation, but will try to use full station names the majority of instances.*

environment --> geographical setting

*-OK, fixed*

.. including station name and station code, latitude .... and the ranges of the data records used in this study.

*-Fixed*

Then add here:

For brevity, we often refer to stations below by their station codes rather than their full names e.g. QB for Quarry Bay.

*-Done.*

You use station codes a lot in the text which seems unnecessary to me when it would be much clearer to the reader to use the place names. There is no space shortage here. So, if you were to remove them all and replace with the names then the above sentence would not be needed.

*-We decided to keep the station codes on the first figure and tables, but will try to spell out the place name in all other locations.*

133-137 - I know what you mean here but this sentence is rather a mouthful. Can you split into two? Also the sentence at line 143 'The tidal potential' should come earlier.

*-Done, and done!*

- 'effects are eliminated'. Why? I don't understand this. They would be eliminated only in an analysis of 18.6 years and would well and truly still be present in a 1-year analysis.

*-Sorry for the confusing statement. We have changed it to better explain the process as:*

"Nodal variabilities are typically present with similar strengths in both the observed tidal record and in the tidal potential. Therefore, when the observed data (harmonically analyzed in one-year windows) is divided by the potential (also analyzed in one year windows, nodal effects are mostly cancelled in the resulting admittance time series."

*-We have also moved this statement down to come after the definition of admittance so it makes more sense.*

- 'may not always hold true'. You could refer to Amin's papers.

*-I added the citation for Amin here.*

start a new para at 'The result'

-Done

-.. analysis window (e.g. at mid-year) ..

*-Done*

- more apparent in the data sets used here

*-Fixed*

- ... MSL variability (Appendix 1). With the use of the TACs we determine ...

*-Fixed*

.. (delta-HAT) (see Appendix 1).

Start a new para.

*-OK, done.*

- mmm should be mm m

*-Fixed, sorry about the careless mistake!*

- The use of a window of a year in a harmonic analysis

*-OK*

181-183 This sentence could come earlier where you mention the Appendix

*-I moved it up and changed the text there slightly*

start new para at 'For the'

*-Done*

Here you start using codes in the text. I have no idea what TPK means and I can't be bothered referring all the time to Figure 1

*-After re-reading closely, I can see your point. We spell out the full station names now.*

years should be 31 (1986-2016)

*-Fixed*

.. determinations (Table 1).

*-OK*

.. in the TAC values over time .. [although I am not sure I understand this. I would reword this and simply say that as the TAC could change over time you have adopted a common epoch for the work]

*-Fixed*

twiddle 12-30 --> 12-31 (Table 1).

*-Fixed*

- I was not provided with the supplement

*-Sorry, I thought it was uploaded, but I will instead add an appendix to the main text.*

- I believe Victoria Harbour is usually spelt with a 'u'. It would be good to show it on Figure 1.

*-You are right, HK uses British English, and I knew that because I live here and can see the harbor from my rooftop. But I think I over-applied the American English standard in the text.*

.. all other gauges except .. moderately negative ...

*-OK, fixed.*

- you add the acronyms in the header at line 200 so do the same in the header here

*-OK, thanks!*

discrete --> particular

*-OK*

drop 'In Hong Kong' Five stations ...

- HK --> Hong Kong.

*-Sorry, force of habit from living here in "HK"*

- drop 'and we report ..'. Irrelevant.

*-OK, thanks, fixed!*

- OT --> overtides (OT)

*-This is now irrelevant due to the removal of Figure 9(c) at the Edtor's suggestion.*

drop 'an additional'

*-Dropped.*

241-242 reword: .. Therefore, all MSL values reported here are given relative to the HKPD for the epoch 1965-1985.

*-Fixed*

drop 'drastically'

*-OK*

correlated to --> correlated with

*-OK*

though --> although

*-OK*

- I don't understand the sentence 'The TACs'. What does it mean that they are present? You mean they are large or what? And you don't actually show delta-HATs but you do show the TACs of the approximate delta-HATs. This all needs rewording.

*-I think I have fixed it to be clearer, I hope you agree!*

The spatial similarity in the ..

*-OK, fixed*

- with processes at other frequencies, such as at

*-Fixed.*

varies with

*-OK, fixed.*

with the spring-neap

*-Fixed*

drop 'away'

*-We refined this sentence.*

forcing of the tides

*-OK, fixed*

- some records are of shorter length and/or have many gaps, making ..

*-Fixed*

as was

*-Fixed*

perform --> employ

*-OK, fixed*

three-dimensional numerical ocean models to simulate the changing impacts on

*-OK, fixed*

comma before 'to better'

*-OK, fixed*

As I mentioned last time, one limitation of this study is the possibility of instrumental changes in the tide gauges. You don't even mention what sort of gauges they are or what changes there might have been.

*-Thanks for this comment!  This is important to explain, and we apologize for not doing it before!  We have, fortunately, a good working relationship with the relevant authorities at the Hong Kong Observatory, and after a request for information and a short delay, they led us to a webpage on their site that gives the official yearly government reports about all Hong Kong tide gauges, their history, locations, and instruments (https://www.hko.gov.hk/publica/pubsmo.htm). Four of six of the HKO gauges (Quarry Bay, Tai Po Kau, Tsim Bei Tsui, and Waglan Island) are sea level pressure transducer types of gauges, and the other two (Shek Pik and Tai Miu Wan) are pneumatic type tide gauges.  The Quarry Bay gauge was updated from a float type gauge recently (2017), and the Tai Po Kau gauge was also updated from a float gauge in 2006.  Neither of these times correspond to any obvious anomalies in the tidal admittance records (the large changes at Tai Po Kau predate this by a few years), so we conclude that the instrumental changes were not a factor in the observed variability.  Finally, all gauges operated by the HK Marine Department were all set up in 2004 as sea level pressure transducers.  We now give this information in the Methods (Sec 2.1), and reprise the gauge changes in the Discussion (Sect 4.3).*

drop hyphen in sea level. There should be a hypen only when used as an adjective e.g. sea-level rise.

*-Fixed here and elsewhere.*

can be positively reinforced by what? by MSL changes?

*- We have changed this sentence to read:*

"The δ-HATs and D1/D2 TACs results illustrate that the tidal variability of multiple constituents  can may be positively additive, and may reinforce MSL changesd at some locations"

agitate --> aggrevate

*-Fixed*

360-364 these web addresses should have http or https

*-Fixed*

interest --> interests

*-Fixed*

383/422 - you could just call it SCS if has been defined in the text

*-We decided to spell it out since I don't refer to it too often in the revised manuscript*

figure 1/2 - add extra names as mentioned above

*-Done*

figure 2 - red on dark blue is not good. I would add China.

*-We tried a different color scheme and increased the fonts*

Taiwan Strait and Luzon are very small and unreadable when printed in A4

*-Fonts are increased and some labels moved to be clearer*

Figure 3 etc. - mention again that red/blue is +/-.

*-OK*

Black marks indicate TACs which are not significantly different from zero.

*-OK*

figure 7 - as I understand it this is not delta-HAT but the TAC computed from the approximate delta-HAt time series made from 4 constituents. Right?

*-Yes, and we say this better in the caption now.*

Please reword the caption to make that clear and not read as jargon. Also you have not defined what delta-HAT4 means in the text.

*-Thanks, we admit that this caption didn't read smoothly. We have attempted to spell this out better as:*

"**Figure 7** The tidal anomaly correlation computed from the combination of the four largest tidal constituent amplitudes (given by the detrended sum of the $M_2 + S_2 + K_1 + O_1$) as a proxy for the change in the approximate highest astronomical tide ($\delta$-HAT) relative to detrended MSL in Hong Kong, with the marker size showing the relative magnitude according to the legend, in units of mm m$^{-1}$. Red/blue markers indicate positive/negative TACs, and black markers indicate TACs which are not significantly different from zero. "

*-The delta-HAT$_4$ was a mistake on the figures and has been cleared up*

sea-level --> sea level

*-Fixed here and elsewhere*

giving the station names and station codes, ... year of the available records, as well as the range of data analysed ..

-Fixed

- and O1 over the period 1986-2016.

*-Fixed*

- TACs over the period 1986-2016.

*-Fixed*

TAC --> TACs

*-Fixed*

Referee 2

L99: mmm-1 is quite confusing; please add a space where appropriate.

*-We have fixed this here and elsewhere, it was a careless mistake.*

LL162: what is delta-hat? On L162 if seems to be "change in the highest astronomical tide ($\delta$-HAT)", whereas on L163-164, we're told that "…the full tidal range ($\delta$-HAT)." I assume it is the astronomical tide, but this could be clearer.

*-Reviewer #1 had the same comments, and was more insistent on this being explained better, so we have added an Appendix that I hope clear up the confusion!*

Forwarded comment

John - you will have had my review through the system. I didn't pick up on them not resolving the overtides as xxx commented on. See below. I don't know if he will reply to you himself.

"Re:Devlin... I just glanced at it. I noticed that they didn't really learn anything from my comment. Figure 9 still has annual estimates of tides like MO3 which are affected by close frequencies, but now added together with a bunch of other compound tides, perhaps in the hope that if you add together a bunch of dubious time series you'll end up with something legit."

*-Thanks for the comments.  I realize now that the OT plot in Fig 9(c) doesn't add much information, and only confuses things.  So, I have removed it and only kept the other three parts.*

Editor comments.

Regarding Figure 9 and overtides, I am not expert (Referee 1 and the commenter are) but I do know that most of your subscript-4 (and subscript-3?) constituents are from non-linearity, whereas M6 is mainly from friction on the M2 flow. There might even be some forcing from higher-order astronomical terms. So I do doubt the value of figure 9c. You don't conclude much from it and I think any gain in understanding would need grouping of the constituents according to their origin (non-linearity, friction, . .).

*-I have re-done Figure 9 to be only 3 parts, removing the OT plot in 9(c).  I agree that the OT plots should not be there, and I should have removed it last revision since I removed other OT related material.*

Line 310. "multiple tides" -> "multiple tidal constituents"?

*-OK fixed!*

**Final Note to Editor about authorship**

**-**Dr. Huthnance, thank you once again for your help, patience and understanding with our manuscript!  I wanted to add a final note about the authorship of this paper.  Last revision I had mentioned some affiliation changes, and in this (hopefully final) version I want to clarify and confirm these changes for the final publication.

All of this paper's authors have now officially moved to our new positions in China.  So, the affiliation order should be:

-**First**:

Department of Geography and the Environment,

Jiangxi Normal University, Nanchang, Jiangxi, China

**Second**:

Institute of Space and Earth Information Science, The Chinese University of Hong Kong, Shatin, Hong Kong SAR, China

**And third** (only for the first two authors, Devlin and Pan):

Shenzhen Research Institute, The Chinese University of Hong Kong, Shenzhen, Guangdong, China

Please instruct us if there is anything else we need to do to confirm and validate these correct affiliations!

And again, Thank you for all your help!

[revised manuscript text omitted]

---

## Author Response (AR3)

Topic Editor Decision: Publish subject to technical corrections (05 Jun 2019) by John M. Huthnance

Comments to the Author:

Dear Authors

Thank-you very much for your re-revised manuscript and all the attention to the reviewers' comments.

Inevitably on reading it through again I have a few more comments. Please see "Detailed comments" below. These are "Technical corrections" meaning that you should deal with these and then enter the manuscript in the Copernicus/OS publication system directly with no more intervention by myself. There will be copy editing and you should check that the final version keeps you intended meaning.

Thank-you for submitting to Ocean Science.

Yours sincerely

John Huthnance

*Thank you for your help and guidance on this manuscript, Dr. Huthnance! I appreciate everything that you have done for me along the way! I am very glad to make these final changes and finalize this paper!*

Detailed comments.

Lines 28-29. You won't need these!

*-OK, removed*

Line 49. ". . under MSL rise. Overall . ." ?

*-Fixed*

Line 92. Better "Pacific; TAC quantifies . ." ?

*-OK, fixed*

Line 93. Better "Devlin et al., 2017a); they found that . ."?

*-OK, fixed*

Line 100. Better with "," after "statistics"

*-OK, fixed*

Line 103. Better "A recent paper performed a similar analysis in the Atlantic . ." or "A recent paper took a similar analysis approach in the Atlantic . ."

*-OK, fixed*

Lines 143-144. Delete one of the two "all"

*-OK, fixed*

Line 171. ". . seem valid . ." or ". . seem to be valid . ."

*-Oops, thanks for catching that! Fixed now.*

Line 176. ". . which is more apparent . ."

*-Sorry for the careless mistkaes!  Fixed now.*

Line 182. ". . The approximation δ-HAT . ."??

*-OK, fixed*

Line 184. ". . largest tidal constituents (M2, S2, K1, and O1) . ."

*-OK, fixed*

Line 269. ". . The overtides (OT) band . ." (unless you meant to delete this sentence).

*-I did mean to omit this, thanks for catching it!*

Lines 328-329. ". . may lead to larger tides . ." ? [Not clear that the forcing changes].

*-OK, fixed!*

Line 333. "constituents . ."

*-OK, fixed*

Line 355. "quickly" -> "briefly"

*-OK, fixed*

Line 368. "but not as much for the δ-HATs." If you mean that TAC was more significant than δ-HAT change, then better "but δ-HAT changes were less significant". If you mean that δ-HAT change was more significant than TAC, then "but not as significant as the δ-HAT changes."

*-I meant the first one; fixed now!*

Line 373. Omit "may also be a factor"

*-Omitted*

Line 756. Add "," after "codes"

*-OK, fixed!*